# FATE: A Formal Benchmark Series for Frontier Algebra of Multiple Difficulty Levels

**Jiedong Jiang**[1*]  **Wanyi He**[2*]  **Yuefeng Wang**[2*]  **Guoxiong Gao**[2*]  **Yongle Hu**[2]
**Jingting Wang**[2]  **Nailin Guan**[2]  **Peihao Wu**[3]  **Chunbo Dai**[3]  **Liang Xiao**[4†]
**Bin Dong**[567†]

[1]Westlake Institute for Advanced Study, Westlake University  [2]Peking University  [3]Ubiquant  [4]New Cornerstone Science Laboratory, School of Mathematical Sciences, Peking University  [5]Beijing International Center for Mathematical Research and the New Cornerstone Science Laboratory, Peking University  [6]Center for Machine Learning Research, Peking University  [7]Center for Intelligent Computing, Great Bay Institute for Advanced Study, Great Bay University

`jiangjiedong@westlake.edu.cn`  `{2501110011,2501110002,samggx,huyongle,`
`wangjingting2023,nailinguan55}@stu.pku.edu.cn`  `{phwu,cbdai}@ubiquant.com`
`lxiao@bicmr.pku.edu.cn`  `dongbin@math.pku.edu.cn`

## Abstract

Recent advances in large language models (LLMs) have demonstrated impressive capabilities in formal theorem proving, particularly on contest-based mathematical benchmarks like the IMO. However, these contests do not reflect the depth, breadth, and abstraction of modern mathematical research. To bridge this gap, we introduce **FATE** (Formal Algebra Theorem Evaluation)[1], a new benchmark series in formal algebra designed to chart a course toward advanced mathematical reasoning. We present two new components, FATE-H and FATE-X, each with 100 problems in abstract and commutative algebra. The FATE series spans a difficulty spectrum from undergraduate exercises to problems exceeding PhD qualifying exams. Notably, FATE-X is the first formal benchmark to surpass both PhD-level exam difficulty and the coverage of the Mathlib library. Our evaluations of state-of-the-art LLM provers on this new benchmark reveal a stark performance gap compared to contest math: the best model achieves only 3% (pass@64) accuracy on FATE-H and 0% on FATE-X. Our two-stage evaluation reveals that models' natural-language reasoning is notably more accurate than their ability to formalize this reasoning. We systematically classify the common errors that arise during this formalization process. Furthermore, a comparative study shows that a specialized prover can exhibit less effective reflection than general-purpose models, reducing its accuracy at the natural-language stage. We believe FATE provides a robust and challenging benchmark that establishes essential checkpoints on the path toward research-level formal mathematical reasoning.

## 1 Introduction

The emergence of reasoning models (Guo et al., 2025; OpenAI, 2025; Anthropic, 2025; Gemini Team, Google, 2025) has boosted the prospect that AI will assist in frontier mathematical research and produce rigorous proofs. Although modern mathematics has been successfully built upon the foundation of natural language proofs, complex proofs generated by current models remain unreliable. A major obstacle to improvement lies in verification: the more advanced a natural language proof is, the more it relies on limited human experts for rigorous

---

[*]Equal contribution.

[†]Corresponding authors.

[1]The FATE benchmark series is open sourced at `https://github.com/frenzymath/FATE`. The evaluation code is open sourced at `https://github.com/frenzymath/FATE-Eval`.

verification, resulting in a process that is slow, unscalable, and error-prone. This creates a critical bottleneck for iterative development. In contrast, formal verification through proof assistants like Lean (Moura & Ullrich, 2021) provides an automated, scalable, and reliable alternative for proof checking (see Appendix A for an introduction).

Following this direction, the research community has made rapid progress in formal automated theorem proving with large language models (LLMs). State-of-the-art models (Ren et al., 2025; Wang et al., 2025; Lin et al., 2025; Zhou et al., 2025; Chen et al., 2025) have achieved impressive results on existing benchmarks and even on IMO contests. However, current benchmarks have largely focused on two areas that differ significantly from modern mathematical research: contest-style problems (Zheng et al., 2022; Liu et al., 2023) and introductory university-level mathematics (Azerbayev et al., 2022; Tsoukalas et al., 2024).

Mathematical contests often prioritize clever tricks over the systematic application of theoretical frameworks, and introductory university courses operate at a lower level of abstraction. In contrast, advanced mathematics is more open-ended, requiring not only the comprehension and application of broad, deeply nested abstract concepts but also the exploration of new insights and the creation of theoretical frameworks. To address this gap, we introduce two benchmarks: the graduate-level FATE-H (Formal Algebra Theorem Evaluation-Hard) and the PhD qualifying-exam-level FATE-X (-Expert). These benchmarks build upon the undergraduate-level FATE-M benchmark by Shen et al. (2025) in the same domain, forming a progressive series of increasing mathematical difficulty designed to assess formal reasoning from undergraduate to post-PhD qualifying exam levels.

FATE focuses on abstract and commutative algebra, a domain emphasizing abstract proofs and the study of structures, reflecting the character of modern mathematics. All problems were selected by expert mathematicians and formalized by specialists to ensure quality and originality. To our knowledge, FATE-X is the first formal benchmark whose mathematical difficulty exceeds that of PhD qualifying exams and whose formalized content surpasses the current scope of Mathlib (mathlib Community, 2020), the mathematical library of Lean.

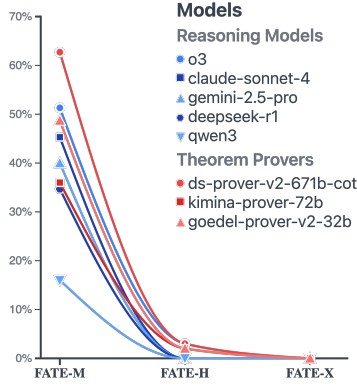

(a) Formalizaton accuracy (Pass@64) across FATE-M/H/X benchmarks.

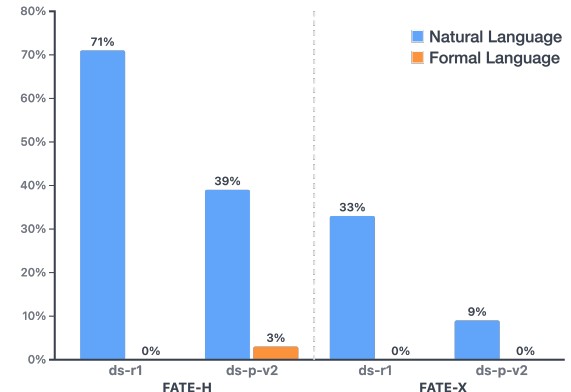

(b) Intermediate natural language (Pass@1) vs. formal language (Pass@64) accuracy on FATE-H/X.

Figure 1: Main experimental results. (a) Formalization accuracy drops sharply along the difficulty progression of FATE-M, FATE-H ($\leq 3\%$), and FATE-X ($0\%$). (b) A significant gap exists between intermediate natural language reasoning and final formal proof generation. Model abbreviations: ds-r1 (DeepSeek-R1), ds-p-v2 (DeepSeek-Prover-V2).

To establish baseline performance, we conducted a comprehensive, two-stage evaluation of state-of-the-art LLMs and specialized provers, mirroring their observed common process of first generating a natural language Chain-of-Thought (CoT) and then formalizing it. We used expert human assessment for the CoT and automated verification for the final Lean code. This dual analysis revealed that formalization accuracy on our research-oriented benchmarks is exceedingly low—with top models achieving just 3% on FATE-H and 0% on FATE-X—a stark drop from their performance on contest-level problems. In contrast,

the natural language reasoning was significantly more accurate, identifying the translation from informal reasoning to formal code as the primary bottleneck. Our case studies of this bottleneck showed that errors related to Mathlib hallucinations and Lean proficiency were the most common, while Misalignment issues were infrequent.

We also compare a general reasoning model (DeepSeek-R1) with a specialized theorem prover (DeepSeek-Prover-V2) to analyze differences in their reasoning behavior. Among our findings, DeepSeek-Prover-V2 exhibits less effective reflection in natural language reasoning, resulting in significantly reduced accuracy at this stage.

To summarize, our core contributions are as follows:

1. **Progressive Benchmark Creation.** We introduce two new benchmarks, FATE-H and FATE-X (100 problems each), and extend the existing FATE-M benchmark from 141 to 150 problems with quality improvements, forming the complete FATE series. This series spans from undergraduate to post-PhD qualifying exam level in algebra. To our best knowledge, *FATE-X is the first formal benchmark to surpass PhD qualifying exam difficulty and Mathlib's formalization coverage.*

2. **Baseline Performance Evaluation.** We comprehensively evaluate state-of-the-art models, establishing performance baselines where the best model achieves only 3% (pass@64) on FATE-H and 0% (pass@64) on FATE-X.

3. **Natural and Formal Two-Stage Output Analysis.** Based on the pattern of natural language reasoning followed by formalization in model outputs, we conduct a detailed two-stage analysis. Our evaluation reveals: (1) significantly higher accuracy in natural language reasoning versus formalization; (2) a classification of common formalization errors: Mathlib hallucinations and Lean proficiency issues were the most common. General capability issues occurred with intermediate frequency, while misalignment were the least common; and (3) in comparative studies, that specialized provers (DeepSeek-Prover-V2) exhibit less effective reflection in natural language reasoning than general-purpose models (DeepSeek-R1), resulting in reduced accuracy at this stage.

## 2 Related Works

**Natural and Formal Mathematical Benchmarks**  Natural language benchmarks by Cobbe et al. (2021) and Hendrycks et al. (2021) cover elementary to high school problems, where leading model performance is nearing saturation. More challenging competition-based benchmarks include those by He et al. (2024), Gao et al. (2025), and Gulati et al. (2024), with progression to university and graduate levels in works by Sawada et al. (2023), Fan et al. (2024), and Chernyshev et al. (2025). Research-level benchmarks by Zhang et al. (2025), Glazer et al. (2024), and partially Phan et al. (2025) represent the current frontier. A common limitation of these benchmarks is their reliance on final-answer verification.

Formal benchmarks enable reliable proof assessment. The miniF2F benchmark (Zheng et al., 2022) focuses on mathematical competitions, with similar efforts including Liu et al. (2023) and Tsoukalas et al. (2024). University-level formalization includes ProofNet (Azerbayev et al., 2022), hybrid datasets (Ren et al., 2025; Yu et al., 2025), specialized combinatorics benchmarks (Xiong et al., 2025; Liu et al., 2025), and the abstract algebra benchmark FATE-M (Shen et al., 2025). Hu et al. (2025) provides undergraduate-accessible problems extracted from formal projects by supplying all necessary lemmas and definitions. These are primarily implemented in Lean, building on Mathlib (mathlib Community, 2020).

**Formal Theorem Proving with Language Models**  Early systems by Polu & Sutskever (2020) pioneered search-based proof generation, followed by methods employing variants of best-first search (Yang et al., 2023; Lin et al., 2024; Wu et al., 2025; Li et al., 2024; Xin et al., 2025) and variants of Monte Carlo tree search (Lample et al., 2022; Gloeckle et al., 2024; Xin et al., 2024). More recently, state-of-the-art provers such as those by Ren et al. (2025); Lin et al. (2025); Wang et al. (2025); Zhou et al. (2025); Chen et al. (2025) have shifted to single-pass generation. Trained with large-scale reinforcement learning, these models produce long

chain-of-thought reasoning to decompose and correct Lean code, achieving accuracy rates approaching 100% on the miniF2F benchmark (Zheng et al., 2022).

# 3 THE FATE BENCHMARK SERIES: DESIGN AND CURATION

The FATE series consists of three benchmarks of increasing mathematical difficulty: FATE-M, FATE-H, and FATE-X. Among these, FATE-H and FATE-X are newly introduced, each containing 100 problems. The existing FATE-M has been expanded from 141 to 150 problems, with improved natural-language comments and formalization details while preserving the original mathematical content.

FATE focuses on abstract and commutative algebra, emphasizing proofs based on abstract properties and the study of algebraic structures instead of equation-solving or numerical computation, thereby reflecting the character of modern mathematics (see Appendix B.2.1 for common problem types). This area is particularly suitable for testing research-level reasoning due to its abstract and self-contained nature, which enables deep problem-solving without requiring extensive mathematical input from other fields. Lean's existing algebra libraries provide a solid foundation for research-level questions, enabling one to attack advanced problems in this area with minimal additional formalization. Unlike benchmarks based on contest problems or introductory undergraduate material, FATE is designed to establish a difficulty progression toward modern mathematical research, structured as follows:

- FATE-M: Textbook-level basic exercises.
- FATE-H: Problems at the level of honors course exams or graduate-level difficulty.
- FATE-X: Problems at the level of PhD qualifying exams or beyond.

This graded structure offers a more comprehensive evaluation than a single accuracy score, preventing strong performance on simpler tasks from masking weaknesses in advanced reasoning, or vice versa.

To demonstrate that the mathematical difficulty is progressive and reaching the PhD qualifying exam level, we first describe the curation process in Section 3.1, then focus on the mathematical content, followed by a description of formalization properties in Section 3.2. Representative examples from each benchmark are provided in Appendix B.1.

## 3.1 BENCHMARK CURATION

**Data Sources**  Mathematical problems for FATE are selected from established sources. The supplementary FATE-M problems are drawn from undergraduate textbooks. FATE-H and FATE-X problems are sourced from: (1) more than 20 standard undergraduate and graduate textbooks, such as Lang (2012) and Eisenbud (2013); (2) publicly available PhD qualifying exams from various universities and exams for honors undergraduate courses; and (3) research papers and advanced community-driven resources such as the Stacks Project (Stacks Project Authors, 2018).

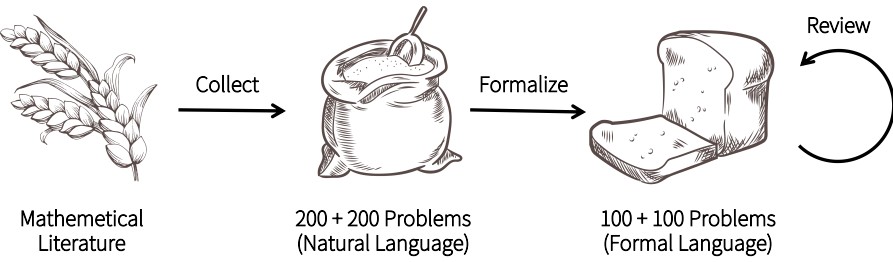

Figure 2: The curation process of FATE-H and FATE-X benchmarks

**Curation Workflow**  The curation process for FATE-H and FATE-X is highly demanding, as it requires experts with deep mathematical knowledge, specialists in Lean formalization, and extremely rare individuals with dual expertise. Figure 2 illustrates the curation process.

1. **Collection**: We collected and classified 400 candidate problems into two difficulty-based categories: FATE-H and FATE-X, each containing 200 problems. The initial curation was conducted through a four-day collaborative effort involving nearly twenty postdocs and PhDs in pure algebra from top institutions, under the supervision of leading researchers. This process ensured the final selection is mathematically profound, well-balanced across key topics, and challenging for both human and machine reasoning.

2. **Formalization**: A subset of 200 problems (100 from each category) was formalized in Lean by five experienced specialists, all contributors to Mathlib, including some involved in the ongoing formalization of Fermat's Last Theorem (Buzzard, 2024). This effort was organized into five dedicated workshops, each involving over five hours of collaboration.

3. **Review**: The formalizations were checked and corrected by two reviewers with strong mathematical backgrounds and Lean expertise, each dedicating over 20 hours. Accuracy was further verified with assistance from external experts in the Lean community.

### 3.2  Benchmark Characteristics

Problem distributions by mathematical topics in FATE-H and FATE-X are provided in Appendix B.2.2 and Figure 3. The charts illustrate a shift in focus from abstract algebra in FATE-H to advanced commutative algebra topics in FATE-X.

**Progressive Difficulty**  Although a consensus exists among mathematicians regarding problem difficulty, it lacks objective measures. We support this from three aspects:

- **Case Studies** We present three examples (one from each benchmark) in Appendix B.3.1 involving similar mathematical topics with increasing difficulty. Analysis shows that these problems require: (1) direct application of a theorem as a linear deduction (FATE-M), (2) synthetic analysis of several direct results as integrative reasoning (FATE-H), and (3) exploration and analysis of new mathematical objects after synthesis as recursive structural analysis (FATE-X). This forms a clear difficulty progression.

- **Human Performance Metrics** We conducted a human experiment to quantitatively test the benchmark difficulty: Each participant, either a Ph.D. student or a postdoctoral researcher in algebra, was randomly assigned 5 non-overlapping natural language problems from FATE-H and FATE-X to solve in 2.5 hours. The participants achieved an accuracy of 73% on FATE-H and 21% on FATE-X, reflecting the significant difficulty progression across the benchmarks. The performance of LLMs also reflects a similar difficulty gradient, as detailed in Appendix B.3.2.

- **Expert Assessment** We conducted a questionnaire survey with ten professors from top institutions specializing in algebra (see Appendix B.4 for details). The results indicate that when comparing FATE to classical textbooks and ProofNet, experts consistently rated FATE significantly higher in difficulty, coverage, and originality. Regarding the suitability of FATE problems for PhD qualifying exams, 7/10 experts indicated that the problems were either directly appropriate for algebra/commutative algebra qualifying exams or notably more challenging than standard PhD qualifying exam questions.

**Formalization Properties**  Due to its scope and difficulty, FATE-X uses mathematical definitions not yet present in Mathlib. In total, 38% of its problems require new definitions, which are formalized before the problem statement. This subset of problems averages 2.4 new definitions each. These include advanced concepts from commutative algebra, such as local complete intersections and Gorenstein rings. Consequently, LLMs must derive the necessary lemmas to use these definitions effectively. Moreover, to solve these challenging problems successfully, models are expected to discover mathematical phenomena, abstract them into useful lemmas, and, when needed, spontaneously formulate new definitions. This capability is critical for research-level mathematical formalization and problem-solving.

All benchmark problems follow strict formalization standards: each Lean file has only one `sorry` after the final theorem; natural language descriptions in LaTeX are included as comments before formal statements; files depend only on Mathlib and are self-contained; and universe levels are fixed to prevent issues from category theory.

## 4  EXPERIMENTS AND RESULTS

In Sections 4.1 and 4.2, we detailed the experimental setup and baseline results on FATE. Observation of the model outputs indicated a two-stage generation pattern, where natural language precedes formalization. Consequently, Sections 4.3 and 4.4 analyze the impact of this two-stage process on the models' final accuracy separately. Section 4.5 presents a comparative study between a general reasoning model and a theorem prover. Finally, Section 4.6 provides a comprehensive discussion synthesizing these findings.

### 4.1  EXPERIMENT SETUP

Our experiment included general reasoning models such as o3 (OpenAI, 2025), Gemini-2.5-Pro (Gemini Team, Google, 2025), Claude-4-Sonnet-thinking (Anthropic, 2025), DeepSeek-R1 (Guo et al., 2025) and Qwen3 (Yang et al., 2025). We also tested state-of-the-art theorem provers, including DeepSeek-Prover-V2-671B (CoT), Kimina-Prover-72B, and Goedel-Prover-72B without self-correction mode. For both categories, we employed whole-proof generation, using pass@64 metric (Chen et al., 2021)-defined as successful if at least one correct proof is found within 64 independent attempts—with a maximum token limit of 64k. For detailed setup and prompt, see Appendices F.1.1 and G.

During formal verification, we implemented multi-process parallelization using the Lean REPL[2] (a read-eval-print-loop for Lean4) to enable efficient large-scale evaluations. Every proof is rigorously checked by the Lean kernel to confirm it contains no `sorry` or compilation errors. Furthermore, to ensure semantic correctness and prevent unintended modifications, we employ string-matching checks to verify the accurate transcription of theorems and definitions. Any model-generated lemmas are also fully compiled to confirm their validity. A detailed methodology is available in Appendix F.1.2.

### 4.2  BENCHMARK PERFORMANCE

Table 1: Formalization accuracy across FATE benchmarks (pass@64, max 64k tokens)

| Model | FATE-M | FATE-H | FATE-X |
|---|---|---|---|
| **Reasoning model** | | | |
| o3 | 51.3% | 3.0% | 0.0% |
| Claude-Sonnet-4 | 45.3% | 0.0% | 0.0% |
| Gemini-2.5-Pro | 40.0% | 0.0% | 0.0% |
| DeepSeek-R1 | 34.7% | 0.0% | 0.0% |
| Qwen3-235B-A22B-Thinking | 16.0% | 0.0% | 0.0% |
| **Theorem prover** | | | |
| DeepSeek-Prover-V2-671B | 62.7% | 3.0% | 0.0% |
| Goedel-Prover-V2-32B | 48.7% | 2.0% | 0.0% |
| Kimina-Prover-72B | 36.0% | 2.0% | 0.0% |

These results reveal a stark contrast in performance: while models achieve reasonable success rate on FATE-M, the best-performing model solves only 3 out of 100 problems in FATE-H, and no model produces any valid Lean proofs in FATE-X. Additional pass@$2^k$ results (for $k = 0, 1, \ldots, 6$) are provided in Appendix F.1.3.

---

[2]https://github.com/leanprover-community/repl

To understand the reason behind low accuracy, we examine the model's output and reasoning content. We observed a consistent behavior across all models with visible reasoning process (DeepSeek-R1 and all theorem provers): even without explicit instructions, models always first work out a full natural language (informal) proof, followed by formalizaion. Among those models without available reasoning content, Gemini-2.5-Pro and Claude-4-sonnet-thinking also tried to solve the problem in natural language first in the output, while only o3 directly output formalization attempts without any natural language content.

Given the models' two-stage process (natural language reasoning followed by formalization), we investigate the impact of each stage on final proof correctness in the following two subsections. For a discussion of the interaction between these stages, see appendix C.1.

### 4.3 Natural Language Reasoning Analysis

In this section, we first present a manual evaluation of the mathematical reasoning in the main experiment's output in Section 4.3.1, which reveals a significant gap between the correctness of the natural language portion and the final formalization. Based on this, we argue that formalization ability is the primary factor in Section 4.3.3. To understand the gap between the intermediate result evaluated by us and the model's real natural language mathematical ability, we also conduct an ablation study in Section 4.3.2.

#### 4.3.1 Manual Evaluation of Natural Language Proofs

The two-stage reasoning phenomenon observed in Section 4.2 prompted us to question: do the models' final formalization failures stem from errors in their initial mathematical reasoning, or from the translation process between natural language and formal language? To answer this, we organized a manual evaluation to independently assess the models' natural language reasoning abilities under the original task settings.

Our mathematics experts evaluated the pass@1 correctness of the natural language proofs generated by DeepSeek-R1, DeepSeek-Prover-V2, Kimina-Prover, and Goedel-Prover-V2 on FATE-H and FATE-X. Manual evaluation criteria and procedures are detailed in Appendix F.2. We attribute errors to four main categories: **Gap**, **Hallucination**, **Reasoning Problem**, and **No Progression**. See Appendix C for definition and examples. The overall results of the evaluation are presented in Table 2. Furthermore, we conducted a pass@16 evaluation for some problems and found that the models do exhibit diverse behaviors on the same problem; see Appendix C for details.

Table 2: Comparing Natural Language (NL) (Pass@1) and Formal Language (FL) (Pass@64) Proof Accuracy on FATE-H and FATE-X

| Model | FATE-H | | FATE-X | |
|---|---|---|---|---|
| | NL | FL | NL | FL |
| DeepSeek-R1 | 71.0% | 0.0% | 33.0% | 0.0% |
| DeepSeek-Prover-V2 | 39.0% | 3.0% | 9.0% | 0.0% |
| Goedel-Prover-V2 | 48.0% | 2.0% | 8.0% | 0.0% |
| Kimina-Prover | 35.0% | 2.0% | 3.0% | 0.0% |

#### 4.3.2 Ablation Study on the Impact of Prompts

Before drawing our conclusions, we must address a potential confounding variable: does the formalization task itself suppress the model's natural language mathematical ability. Consequently, we designed an ablation study testing different prompt settings for DeepSeek-R1, see C.2. We find that the models' natural language mathematical ability is higher when unburdened by the prompt to generate a formal proof. However, modifying the baseline prompt to explicitly require a "math-before-lean" output had almost no impact on accuracy regarding Deepseek-R1.

### 4.3.3 Analysis of Main Findings

1. **Primary Bottleneck: The Translation to Formal Language.** First, for all models tested, intermediate natural language reasoning accuracy (pass@1) far exceeds final formalization accuracy (pass@64). The ablation study further suggests that the models' mathematical ability when tasked only with generating a natural language proof provided with an informal statement is likely even higher than what this evaluation captured. In conjunction with our finding (see Section 4.4) that formalization attempts are highly aligned with their preceding natural language reasoning, these results together show that the bottleneck is not the mathematical ability itself. Rather, the critical challenge lies in the translation and implementation of a correct natural language proof into an absolutely precise formal language.

2. **Difference in Mathematical Abilities Between Models.** At the level of natural language mathematical reasoning, the general-purpose reasoning model behave significantly better than the specialized prover models in Table 2. We will conduct a deeper comparative analysis of this phenomenon in Section 4.5.

### 4.4 Formalization Error Analysis

This subsection classifies and quantifies common LLM formalization errors. Our human Lean experts analyzed proof attempts from DeepSeek-Prover-V2 on DeepSeek-R1 on the FATE-H benchmark, selected for its significant gap between natural language and formal language accuracy. After natural language evaluation in Section 4.3.1, experts manually counted distinct formalization errors in mathematically correct but formally incorrect attempts. These errors fall into four categories:

1. **Mathlib Hallucinations**: Errors in this category involve the generation of non-existent or incorrectly used Lean theorems or definitions;

2. **Lean Proficiency Issues**: These are errors related to a lack of understanding of Lean's specific syntactic rules, sophisticated type system, or idiomatic proof structures;

3. **General Capability Issues**: This category includes problems such as modifying headers and the others -leaving `sorry`s, producing repetitive output or unmatched brackets;

4. **Misalignment**: This error occurs when the model's formal proof is inconsistent with previous mathematical reasoning.

For more details of these four categories, see Appendix D.

Table 3: Formal error counting result for models on FATE-H

| Formal Error | DeepSeek-Prover-V2 | DeepSeek-R1 |
|---|---|---|
| Mathlib Hallucination | 35/39 | 70/71 |
| Lean Proficiency | 36/39 | 70/71 |
| General Capability (header) | –/– | 63/71 |
| General Capability (others) | 19/39 | 18/71 |
| Misalignment | 3/39 | 0/71 |

For results in Table 3, the denominator is the number of whole proofs generated by LLMs, which intermediate natural proof is manually judged to be correct. Among the four errors, Mathlib hallucinations and Lean proficiency issues were the most common errors, frequently recurring within almost every single proof attempt. In contrast, misalignment were remarkably infrequent. This suggests that a retrieval-augmented generation (RAG) system, which retrieves relevant theorems from Mathlib and provides accurate type information, could help improve formalization performance. As DeepSeek-Prover-V2's output lacks headers, we could not include its data for header-related issues. However, among other general capability issues, DeepSeek-Prover-V2 performed notably worse. Additionally, we found that only DeepSeek-Prover-V2 exhibited repetitive proof steps and misalignment.

A case study in Appendix D.2 revealed that the types of formalization errors on FATE-X were consistent with those on FATE-H. Notably, for problems involving new definitions, models rarely generated auxiliary lemmas to aid in proving.

## 4.5 General Models vs. Theorem Provers

As demonstrated by the experimental results in Section 4, the general-purpose reasoning model significantly outperforms the specialized proof-assistant model in terms of natural language success rate. To investigate the root cause of this performance gap, we conducted a detailed qualitative analysis of the reasoning processes of DeepSeek-V3 (Liu et al., 2024), DeepSeek-Prover-V2, and DeepSeek-R1-a series of models all post-trained from the same base model. More detailed experimental setup, results and case studies are provided in Appendix E. Here, we summarize the main conclusions:

In the initial phase of reasoning, all three models often adopt similar strategies or invoke comparable concepts. However, as the reasoning deepens and established problem-solving patterns fail to address the specific problem, their behaviors diverge significantly. The typical behavior for DeepSeek-V3 is to directly assemble associated concepts into a superficial argument. In contrast, both DeepSeek-Prover-V2 and DeepSeek-R1 attempt to provide more detailed discussions.

The essential difference between the latter two lies in a capability we term **effective reflection**: the ability to locate, diagnose and repair flaws. While DeepSeek-R1 can sometimes achieve this, DeepSeek-Prover-V2 is confined to performing formal reflections, such as starting over or a rhetorical shift without a corresponding logical change. Furthermore, our studying discovers non-aligned phenomena during mathematical reasoning unique to DeepSeek-Prover-V2. These behaviors, which also affect their performance, include questioning the correctness of the problem statement itself after a failure and even engaging in conscious cheating behaviors.

The intermediate natural language accuracy results of three models in Table 8 confirm these observations: the DeepSeek-Prover-V2, which offers detail but lacks effective reflection, achieves a final success rate nearly identical to that of the formally reasoning DeepSeek-V3. Meanwhile, DeepSeek-R1, which possesses this capability, significantly outperforms both. Nevertheless, we must note that even DeepSeek-R1's reasoning ability has its limitations when faced with truly complex reasoning environments.

## 4.6 Implications and Discussion

Our experimental results and analyses lead to two points for discussion regarding future research in automated theorem proving:

First, our findings show that whole proof generation models use a two-stage process for formal proving: generating a natural language proof, then a well-aligned formalization, suggesting these stages are largely decoupled. Considering our findings that the intermediate natural language accuracy of DeepSeek-Prover-V2 is lower than that of DeepSeek-R1, further lower than pure natural language accuracy of DeepSeek-R1. Together, these points strongly suggest that an explicitly decoupled approach, developing a natural language prover and a separate autoformalizer, would gain extra improvement.

Next, our comparative experiments reveal that the general-purpose model performs more "effective reflection", a capability crucial to human mathematical reasoning. In contrast, the specialized prover, despite reinforcement learning in the narrower domain of formalization, did not gain the expected boost in mathematical reasoning. Instead, its performance was often hindered by various misaligned behaviors that degraded its reasoning ability relative to the general model. Considering the comparison with its base model (DeepSeek-V3), this raises the question of whether the DeepSeek-Prover-V2's lack of effective reflection is an unintended outcome of its specialized training scheme. This, in turn, leads to a critical challenge for future work: is it possible to design a training methodology that can simultaneously leverage the precise reward signals from formalization while also fostering essential meta-reasoning capabilities such as "effective reflection"?

Although our findings suggest these research directions, rigorously confirming them is beyond the scope of this article and remains a key task for future work.

## 5 Conclusions

To bridge the gap between contest-style mathematics and modern research, we introduced FATE-H and FATE-X, extending the FATE series of formal algebra benchmarks, which features a graded mathematical difficulty structure and a frontier research focus. Our evaluation of state-of-the-art models reveals their significant limitations in conducting formal reasoning in advanced mathematics, with maximum pass rates of just 3% (Pass@64) on FATE-H and 0% on FATE-X. Further analysis indicates that the primary bottleneck lies in the formalization stage, with errors stemming mainly from hallucinations of the formal library and insufficient language proficiency, rather than in natural language mathematical reasoning, as the two stages exhibit a functional decoupling. Furthermore, we found that general-purpose reasoning models outperform their specialized counterparts, due to a more effective "reflection" capability. These findings collectively point to two fundamental research directions: first, explicitly decoupling the task of natural language reasoning from formalization generation; and second, balancing the pursuit of formal accuracy with the cultivation of effective and aligned reasoning capabilities.

### Acknowledgments

This work is supported in part by National Key R&D Program of China grant 2024YFA1014000, the New Cornerstone Investigator Program, and Ubiquant.

We wish to express our sincere appreciation to Yutong Wang for generously providing the code for the Lean verification. We are deeply grateful to the following individuals for their contributions to the selection of mathematical problems for the FATE benchmarks: Kaiyi Chen, Haocheng Fan, Yiqin He, Yongle Hu, Shanxiao Huang, Yudong Liu, Tian Qiu, Yinchong Song, Peihang Wu, Zhenhua Wu, Tianyi Xu, Zhehan Xu, Huanhuan Yu, Huishi Yu, Jiahong Yu, Zhanhao Yu and Xiao Yuan. We gratefully acknowledge the following individuals for their careful assessment of natural language proofs and insightful feedback: Haocheng Fan, Haoda Li, Zhongjin Yan, Chaodong Yang, Yuji Yang, Shun Yin, Huanhuan Yu, Jiahong Yu and Weijun Yuan. We also thank Nailin Guan, Zixun Guo, Yongle Hu, and Jingting Wang for their dedicated work in formalizing mathematical problems. We extend our sincere thanks to Johan Commelin and Christian Merten for their careful external review and corrections to the benchmark problems. We are also indebted to Siddhartha Gadgil, Filippo A. E. Nuccio and Floris van Doorn for their valuable discussions and suggestions.

Finally, we express our deepest appreciation to the Lean community for developing and maintaining Mathlib—without this foundational infrastructure, this work would not be possible.

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

## A  INTRODUCTION TO FORMALIZATION AND LEAN

**Mathematical Formalization**   Mathematical formalization is the process of translating intuitive, informal ideas and concepts into a precise, unambiguous language defined by strict logical rules. To support this process, *interactive theorem provers (ITPs)* assist users in constructing formal proofs. ITPs typically provide a tactic mode, in which the prover tracks available hypotheses and the current goal as the proof state. Users manipulate this state through concise, high-level commands called *tactics*, and the prover automatically generates corresponding low-level proof code once all goals are solved. Tactics are designed to mirror patterns in natural language proofs, improving both readability and writability. Almost all LLM-based formal theorem provers generate such tactics to construct formal proofs.

Various ITPs are built on distinct but equally sound logical foundations: Metamath (Megill & Wheeler, 2019) and Mizar (Bancerek et al., 2015; Grabowski et al., 2010) are based on set theory; HOL4 (Slind & Norrish, 2008), HOL Light (Harrison, 2009), and Isabelle/HOL (Nipkow et al., 2002) are based on simple type theory; and Coq (Barras et al., 1999) and Lean (Moura & Ullrich, 2021) are based on dependent type theory. The mathematical library available in each prover determines the difficulty of formalizing different topics. Lean's extensive and unified Mathlib (mathlib Community, 2020) distinguishes it from other ITPs.

**Lean and Mathlib**   Lean 4 (Moura & Ullrich, 2021) is an interactive theorem prover based on dependent type theory, with proof irrelevance and non-cumulative universes (Carneiro, 2019). An introduction to the language can be found in (Avigad et al., 2021). Lean 4 is designed to be highly extensible through its metaprogramming capabilities, enabling the accurate and concise extraction of important metadata.

Mathlib4 (mathlib Community, 2020) is a community-driven effort to build a unified library of mathematics in Lean 4. It formalizes a substantial portion of modern mathematics and currently contains over 1,700,000 lines of code contributed by more than 550 developers. Mathlib covers a wide range of topics, including algebraic geometry, analysis, category theory, combinatorics, geometry, general algebra, linear algebra, logic, number theory, order theory, probability theory, and topology. Among these, the formalization in algebra is the most developed. This makes Lean particularly suitable for formalizing advanced mathematics, especially in algebra. Such formalization remains inaccessible in many other interactive theorem provers, which lack similarly extensive foundational libraries.

# B   Benchmark Quality Analysis

This appendix presents the quality of the FATE benchmarks, emphasizing their mathematical content. Following illustrative examples from each benchmark in Appendix B.1, we provide case studies and statistics to demonstrate the diversity (see Appendix B.2) and progressive difficulty (see Appendix B.3) of the problems. The results of an expert survey assessing the benchmarks are reported in Appendix B.4. For comparison, examples from the FATE-M benchmark introduced by Shen et al. (2025) are included. For analyses focused purely on mathematical content, we provide only the natural-language problem statements without formalization details for brevity.

## B.1   Representative Examples

FATE-M

```
import Mathlib

/--
Suppose $D$ is integral domain, $m$ and $n$ are coprime positive integers. Prove
    that for any $a, b \in D$, if $a^{m}=b^{m}$ and $a^{n}=b^{n}$, we have $a=b$.
-/
theorem eq_of_pow_eq_of_coPrime {R : Type*} [Ring R] [IsDomain R] (a b : R)
    (m n : ℕ) (hm : m > 0) (hn : n > 0) (hmn : m.Coprime n)
    (h₁ : a ^ m = b ^ m) (h₂ : a ^ n = b ^ n) : a = b := by
  sorry
```

FATE-H

```
import Mathlib

open IntermediateField RatFunc

/--
Let $\mathbb{F}_4$ be the field with $4$ elements, $t$ a transcendental over $
    \mathbb{F}_4$, and $F =\mathbb{F}_4(t^4 + t)$ and $K =\mathbb{F}_4(t)$. Show
    that $K$ is Galois over $F$.
-/
theorem isGalois_galoisField_adjoin_X_pow_four_add_X :
    IsGalois (GaloisField 2 2)[(X ^ 4 + X : RatFunc (GaloisField 2 2))]
    (RatFunc (GaloisField 2 2)) := by
  sorry
```

FATE-X

```
import Mathlib

/--
Let \( A \) be a domain and \( K \) its field of fractions.
\( x \in K \) is called almost integral if there exists an element
\( r\in A, r \ne 0 \) such that \( rx^n \in A \) for all \( n \ge 0 \).
-/
def IsAlmostIntegral {A : Type} [CommRing A] [IsDomain A] (x : FractionRing A) :
    Prop :=
  ∃ r : A, r ≠ 0 ∧ ∀ n : ℕ, ∃ y : A,
    r • (x ^ n) = algebraMap A (FractionRing A) y

/--
\( A \) is called completely integrally closed if
every almost integral element of \( K \) is contained in \( A \).
```

```
-/
def IsCompletelyIntegrallyClosed (A : Type) [CommRing A] [IsDomain A] : Prop :=
    ∀ x : FractionRing A, IsAlmostIntegral x → ∃ y : A, x = algebraMap A
    (FractionRing A) y

/--
Let \( A \) be a domain. Show that if \( A \) is completely
integrally closed, so is \( A[X] \).
-/
theorem completely_integrally_closed_polynomial_ring {A : Type} [CommRing A]
    [IsDomain A] (h : IsCompletelyIntegrallyClosed A) :
    IsCompletelyIntegrallyClosed (Polynomial A) := by
  sorry
```

## B.2  Diversity

Appendix B.2.1 presents three principal problem types within the series, each accompanied by a representative case. The statistical distribution of mathematical domains is detailed in Appendix B.2.2.

### B.2.1  Problem Types

The FATE benchmark includes a variety of problem types, with three representative categories outlined below.

**Abstract Reasoning Problems**  These problems start from an abstract condition and require proving a general conclusion.

> **(FATE-M)** Prove that if a finite abelian group has order a power of a prime $p$, then the order of every element in the group is a power of $p$.

**Concrete Example Problems**  These problems start from a specific, concrete example and ask for the proof of one of its properties.

> **(FATE-H)** Prove that the order of $\mathrm{Aut}(\mathbb{Z}_3 \times \mathbb{Z}_9)$ is 108.

**Open Constructive Problems**  These problems require the construction of an object that satisfies a specific property.

> **(FATE-X)** There exists two commutative rings $R, S$, such that $R[x]$ is isomorphic to $S[x]$ but $R$ is not isomorphic to $S$.

The first category, abstract reasoning problems, is the most common type in FATE-H and FATE-X. It is noteworthy that the abstractness of a problem type does not directly correlate with its mathematical difficulty. A detailed case study on the benchmark's difficulty gradient is available in Appendix B.3.1.

### B.2.2  Domain Statistics

We classify abstract algebra and commutative algebra into subfields and further break down these subfields into specific topics. For a complete list of subfields, topics, and corresponding statistics, see Table 4; for a visual sunburst chart representation, refer to Figure 3. The chart illustrates a shift in emphasis from abstract algebra in FATE-H to advanced commutative algebra topics in FATE-X.

Table 4: Problem Distribution by Mathematical Topic in FATE-H and FATE-X

| Subfield | Topic | FATE-H | FATE-X |
|---|---|---|---|
| **Abstract Algebra** | | | |
| Group Theory | Basic Axioms and Examples | 4 | 0 |
| | Subgroups and Quotient groups | 3 | 3 |
| | Group Actions and Sylow theorems | 9 | 5 |
| | p-Groups, Nilpotent Groups and Solvable Groups | 2 | 1 |
| Ring Theory | Basic Definitions and Examples | 2 | 3 |
| | PID, ED and UFD | 1 | 4 |
| | Polynomials | 8 | 3 |
| Field Theory | Field Extensions | 11 | 1 |
| | Galois theory | 11 | 11 |
| | Explicit Computations | 11 | 1 |
| Other Topics | Linear Algebra | 3 | 0 |
| | Elementary Number Theory | 1 | 0 |
| **Commutative Algebra** | | | |
| Ideal Theory | Ideals and Modules | 5 | 9 |
| | Localization and Decomposition of Ideals | 3 | 3 |
| | Integral Dependence and the Nullstellensatz | 7 | 5 |
| | Noetherian rings and Chain Conditions | 1 | 3 |
| | Dedekind Domains and DVRs | 4 | 1 |
| | Tensor Product and Flatness | 2 | 8 |
| Dimension Theory and Smoothness | Heights and Dimensions | 2 | 6 |
| | Regular Sequences and Regular Local Rings | 4 | 5 |
| | Smoothness and the Module of Differentials | 0 | 4 |
| | Depth, Cohen-Macaulay Rings and Gorenstein Rings | 0 | 8 |
| Other Topics | Completions and Hensel's lemma | 1 | 4 |
| | Algebraic Number Theory and Valuation Theory | 5 | 3 |
| | Algebraic and Arithmetic Dynamics | 0 | 6 |
| | Homological Methods | 0 | 2 |
| | Noncommutative Algebras | 0 | 1 |

## B.3 PROGRESSIVE DIFFICULTY

Appendix B.3.1 presents three topic-related examples of increasing difficulty, demonstrating the progressive structure of the benchmarks. Furthermore, Appendix B.3.2 provides statistics on model output token length and natural language accuracy to reflect problem difficulty.

### B.3.1 CASE STUDY: A GRADED CHALLENGE ON NON-SIMPLE GROUPS

To demonstrate the difficulty gradient of the FATE benchmark, we analyze the solutions and reasoning contexts generated by DeepSeek-R1 on three topic-related problems from FATE-M, H, and X. As illustrated in Figure 4, the model's generated proofs exhibit a clear progression in structure and complexity, reflecting the increasing mathematical difficulty of the problems.

For the M-level problem, the approach is direct: the model applies Sylow's theorem to a single prime $p$ (as suggested by the problem setup) and reaches the conclusion directly using the given conditions. In contrast, for the H-level problem, the single-prime approach is insufficient. The model synthesizes information from Sylow's theorem applied to all prime factors of the group order and employs a counting argument to derive a contradiction. The X-level problem requires an additional step: after performing the aforementioned analyses, several possibilities remain. To resolve these, one must examine new objects and conduct further

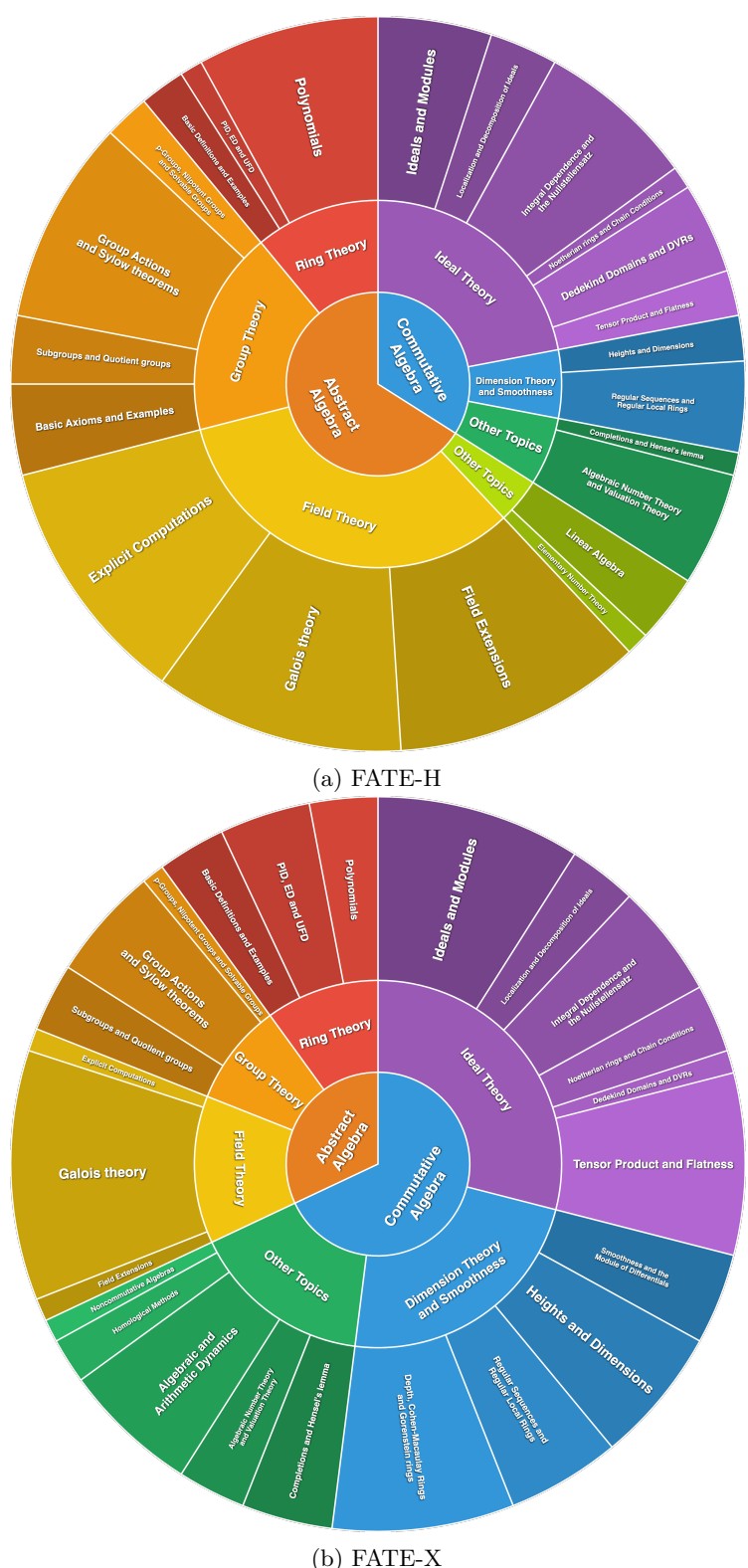

(a) FATE-H

(b) FATE-X

Figure 3: Problem distributions by mathematical topics in FATE-H/X.

analysis using Sylow's theorem—specifically, by investigating the relationship between the normalizer of the Sylow 17-subgroups and the Sylow 3-subgroups.

| FATE-M | FATE-H | FATE-X |
|---|---|---|
| Let $G$ be a finite group of order $p^2 q$, where $p$ and $q$ are primes with $p > q$. Then the Sylow $p$-subgroup of $G$ is normal. (As a consequence, $G$ is not simple.) | Prove that if $\#G = 56$ then $G$ is not simple. | Prove that if $\#G = 1785$ then $G$ is not simple. |
| ```### Approach`
`1. **Sylow's Third Theorem**:`
`<…>`
`2. **Divisors Analysis**:`
`<…>`
`3. **Contradiction for \(n_p = q\)**:`
`<…>`
`leading to a contradiction.``` | ```We consider the Sylow 7-subgroups.`
`<…> By Sylow's third theorem:`
`<…>apply sylow for p=7…>`
`Now consider the Sylow 2-subgroups.`
`<…>apply sylow for p=2…>`
`Therefore, we are left with the case:`
`n_7=8 and n_2=7.`
`<…>`
`We have 48 elements of order 7, so`
`the remaining 56-48=8 elements must`
`form the Sylow 2-subgroup?`
`But wait, we have 7 Sylow 2-`
`subgroups, each of order 8. How can`
`we have only 8 elements left?`
`<…>`
`Thus, G is not simple.``` | ```We will consider the Sylow subgroups`
`for the primes 3,5,7,17.`
`<…>apply sylow for every possible p…>`
` Thus:`
`    n_3 ∈ {7,85,595}`
`    n_5 ∈ {21,51}`
`    n_7 ∈ {15,85}`
`    n_17 ∈ {35}`
`<…>`
`So total nonidentity elements of`
`prime order: 2*n3 + 4*n5 + 6*n7 +`
`560`
`<…long reasoning about counting…>`
`But we have to find a contradiction`
`in the remaining 8 cases.`
`<…>`
`Let's take a specific combination.`
`For example, combination 1: n3=7,`
`n5=21, n7=15.`
`<…long exploration…>`
`But a Sylow 3-subgroup that is`
`normalized by P would give a`
`subgroup generated by P and that`
`Sylow 3-subgroup, which would have`
`order 17*3=51. And we know there are`
`subgroups of order 51. (This is the`
`critic construction)`
`<…long reasoning, failed at the`
`end…>``` |

Figure 4: Proof structure progression generated by DeepSeek-R1 on FATE-M, FATE-H, and FATE-X benchmarks. The colored segments represent different reasoning phases: initial theorem application (blue), synthesis of intermediate results (orange), and exploratory steps (green).

**From Proof Structure to Capability Gradient**  The increasing complexity of the required proof structure is directly reflected in the model's output. The reasoning content produced by the model aligns with this structural progression, as visualized in Figure 4 using distinct colors to mark different reasoning stages:

- The M-level problem exhibits a simple sequential dependency chain, requiring only **linear deduction**. Its monolithic, unidirectional reasoning is represented by a single blue block;

- The H-level problem requires a tree-like dependency structure. The model's initial parallel explorations (blue block) converge into a final synthetic conclusion (orange block)—a process we term **integrative reasoning**;

- The X-level problem corresponds to a more complex, nested proof structure. After initial analysis proves insufficient, one must **extract a new mathematical object** from the proof state and reapply major theorems to it (green block). This nested re-analysis of intermediate results requires **recursive structural analysis**.

In summary, the difficulty gradient in this case arises from the deliberate design of increasingly complex proof structures, which in turn demand a progression of capabilities: from linear to integrative to recursive structural reasoning.

### B.3.2   Auxiliary Statistics on Proof Difficulty

In this section, we present two statistics: (1) human evaluation of the intermediate natural language accuracy during the formalization process of DeepSeek-R1 on FATE-M, FATE-H,

and FATE-X; and (2) the average output token length of various reasoning models when solving natural language mathematical problems without formalization.

As noted at the end of Section 4.2, the model first produces a complete natural language solution before formalizing it. The data in Table 5 reflect expert evaluations of these intermediate natural language solutions (Pass@1), showing a clear decreasing accuracy trend from FATE-M to FATE-H to FATE-X.

Table 5: Intermediate Natural Language Accuracy Across FATE-M/H/X

| Model | FATE-M | FATE-H | FATE-X |
|-------|--------|--------|--------|
| DeepSeek-R1 | 94.7% | 71.0% | 33.0% |

It should be noted that these results do not directly represent natural language performance, since the model's final task is formalization rather than natural language proof generation. An ablation study in Appendix C.2 compares natural language accuracy between a direct natural language task and the intermediate natural language output within the formalization task. The results show that although a pure natural language task yields higher accuracy, the improvement is significantly smaller than the gap between different levels and therefore does not affect the observed decreasing accuracy trend across the three benchmarks.

Table 6: Average Output Token Length on FATE-M/H/X (Pass@1)

| Model | FATE-M | FATE-H | FATE-X |
|-------|--------|--------|--------|
| o3 | 1122.4 | 3799.5 | 6203.5 |
| Claude-Sonnet-4 | 3827.8 | 11872.2 | 12118.8 |
| Gemini-2.5-Pro | 4867.5 | 8508.6 | 14694.1 |
| DeepSeek-R1 | 3845.0 | 10625.1 | 16597.4 |

The token length data in Table 6 indicate a clear increasing trend in output token length from FATE-M to FATE-H to FATE-X across all models.

### B.4 Expert Assessment

We distributed a questionnaire to 10 professors specializing in algebra and number theory at top institutions, consisting of two parts: (1) a comparison of the difficulty of FATE against a classical textbook and the ProofNet (Azerbayev et al., 2022) benchmark, and (2) an assessment of the suitability of FATE benchmark problems for use in PhD qualifying examinations.

In the first part, we divided the FATE-H and FATE-X benchmarks, classical algebra textbooks (Dummit et al., 2004), and the algebra component of ProofNet by mathematical topic. We then randomly sampled 10 problems with balanced topic distribution from each to form three anonymous problem sets. Participants were asked to compare and select the best set across several dimensions. The results are summarized as follows:

- For the most difficult, 10/10 choose FATE.
- For the best coverage, 8/10 choose FATE, 2/10 choose hard to distinguish.
- For the best to test deep understanding, 9/10 choose FATE, 1/10 choose classical textbook.
- For the most original, 9/10 choose FATE, 1/10 choose hard to distinguish.
- For the best to evaluate ability in mathematical research, 9/10 choose FATE, 1/10 choose classical textbook.

It has shown that an absolute majority of experts support the view that the problem set chosen from FATE has the best difficulty, coverage, ability to test deep understanding,

originality, and effectiveness for evaluating ability in mathematical research, compared to the problem sets chosen from classical textbook and ProofNet.

In the second part, we randomly sampled an additional 30 non-overlapping problems from FATE-X while maintaining balanced topic distributions. Minor supplements from FATE-H were used to adjust the proportion of abstract algebra and commutative algebra. Participants were then asked to evaluate the suitability of these problems for PhD qualifying exams based on several criteria. The results are summarized as follows:

- For coverage of knowledge points in PhD qualifying exams, 4/10 choose completely covered, 2/10 choose almost completely covered, 3/10 choose mostly covered with no important points omitted, 1/10 choose important points omitted.
- For difficulty, 7/10 choose meets or exceeds PhD qualifying exam level, 3/10 choose meets excellent undergraduate level.
- For ability to differentiate students, 3/10 choose very well, 4/10 choose reasonably well, 3/10 choose insufficient.
- For possibility to construct PhD qualifying exam, 4/10 choose possible, 6/10 choose need adjustment. Among those choosing need adjustment, 3/6 report too difficult, 3/6 report need more topics like group representation theory.

The results show that an absolute majority recognize that the problem set meets or exceeds PhD qualifying exam difficulty and mostly covers knowledge points in abstract and commutative algebra with no important points omitted. One important factor affecting both differentiation ability and exam construction is that the problem set is more difficult than standard PhD qualifying exams.

## C  Analysis of Natural Language Output

### C.1  Interaction Between Natural Language Reasoning and Formalization

In the main body of this paper, we conclude that the natural language mathematical reasoning stage and the formalization stage are functionally decoupled in current models. This appendix serves to explore the few, yet insightful, cases where a positive interaction between these two stages does occur.

A clear example of this interaction is when the formalization process reveals a **mathematical error** in the preceding natural language proof: In the following example, the model DeepSeek-R1 initially derived an incorrect formula for an element count. It was only when attempting to formalize this flawed formula that the model recognized its logical mistake and reverted to the natural language stage to correct it.

**Output of DeepSeek-R1 (FATE-X)**

```
<...>
total_elements_accounted = 1 + 2n_3 + 4n_5 + 6n_7 + 560.
// The model initially provides a counting formula, but forgets to include
    elements of composite order.
<...>
let elements_prime_powers : ℕ :=
      (Finset.univ.filter (fun x : G => ∃ p, p.Prime ∧ orderOf x = p)).card

But note: we are counting elements of prime order?
// During formalization, it reflects on the true meaning of the equation.
<...>
So the total number of elements is:

1 + [n3*2 + n5*4 + n7*6 + n17*16] + (# composite order elements) = 1785.
// It then corrects the counting formula to include all elements.
<...>
```

The comments after `//` in the code block explain the model's behavior.

In addition to this, a similar interaction is observed when a model deems a mathematically valid natural language proof to be **ill-suited for formalization**. In such instances, the model may also abandon its initial natural language proof and return to the reasoning stage to devise an alternative strategy that is more amenable to formalization.

Paradoxically, this consistent pattern of interaction reinforces the hypothesis of functional decoupling. Whether the feedback from the formal language stage reveals a mathematical error or a practical formalization obstacle, the model's response is the same: it **reverts to the natural language stage** to perform the core logical revision. The model does not fix the mathematical logic within the formal framework; instead, it treats the formal language stage as a verifier that sends the task back to a functionally separate natural language "prover". This workflow highlights a clear separation of concerns, further supporting the view that a decoupled, two-stage research approach is a promising path forward.

## C.2 Ablation Study of Prompts

To understand how the prompt influence the model's natural language performance, we designed a pass@1 ablation study testing four different prompt settings for DeepSeek-R1 on FATE-X, see Appendix G.2 for explicit prompts:

1. **Baseline Prompt:** Requires generating a Lean proof and provides the formal statement.
2. **Pure Math Prompt:** Does not require generating Lean and does not provide the formal statement.
3. **Math Output Prompt:** Does not require generating Lean but provides the formal statement.
4. **Math-before-Lean Output Prompt:** Builds on the Baseline Prompt, adding a clear instruction to "output the mathematical proof first".

Table 7: Natural language accuracy of DeepSeek-R1 under different prompts on FATE-X

|  | Baseline | Pure Math | Math Output | Math-before-Lean Output |
|---|---|---|---|---|
| Accuracy | 33% | 43% | 39% | 32% |

The results, shown in Table 7, reveal the following:

- **The primary influencing factor is the formalization task itself:** When the model's objective includes generating Lean code (comparing Prompt 1 and 3), the correctness of its natural language reasoning significantly decreases.

- **A minor factor is the formal context:** When the model is shown the formal statement (comparing Prompt 2 and 3), the correctness of its mathematical reasoning slightly decreases.

- **Explicitly requiring a math-first output has almost no effect:** The results for the Explicit Instruction Prompt 4 were nearly identical to those for the Baseline Prompt 1.

## C.3 Case Study: Common Errors in Natural Language Reasoning

To better understand the models' weaknesses in reasoning, we analyzed and categorized errors in their natural language proofs. These categories are not a non-overlapping classification but a summary of typical failure modes observed by our experts that covers most issues. Since advanced models often make subtle errors, this classification relies not just on the raw output, but also on an expert's deep understanding of the problem and experience on grading.

### C.3.1 Gap

A **Gap** occurs when the model's reasoning direction is correct but leaves a key, non-trivial step not proved. This often involves secretly assuming a necessary lemma or citing an overly powerful theorem to bypass the argument.

**Natural Language Problem Statement**

> **(FATE-X)** Prove that if $\#G = 396$ then $G$ is not simple.

**Model Output (DeepSeek-Prover-V2)**

```
<...>
   - If $n_2 = 99$, count elements:
     - $99 \cdot 3 = 297$ elements of order $2$ or $4$.
     - Total elements $\geq 120 + 176 + 297 = 593 > 396$, contradiction.
<...>
```

**Analysis**  A standard solution here is to derive a contradiction by counting elements in all Sylow subgroups. However, a crucial premise for this method is the proof that any two distinct Sylow subgroups have a trivial intersection. DeepSeek-Prover-V2 skips this key argument, which is itself a difficult and key part of the problem.

### C.3.2 Hallucination

A **Hallucination** is an explicit statement of a false mathematical fact, typically stemming from an incorrect recollection of a theorem or flawed mathematical intuition.

**Natural Language Problem Statement**

> **(FATE-X)** Let $k$ be a field, and set $A = k[X, Y, Z]/(X^2 - Y^2, Y^2 - Z^2, XY, YZ, ZX)$. Show that $A$ is Gorenstein.

**Model Output (DeepSeek-R1)**

```
<...> For an Artinian local ring to be Gorenstein, it suffices to show that the
socle \operatorname{Soc}(A) = \operatorname{ann}_A(\mathfrak{m}) has \dim_k
\operatorname{Soc}(A) = 1. <...>
```

**Analysis**  This is a factual error; the condition that the socle has dimension 1 is not sufficient to be Gorenstein (the correction need an extra condition). Notably, all tested results (including DeepSeek-Prover-V2, DeepSeek-R1 in 4 attempts) made the exact same mistake on this problem, suggesting a common knowledge gap in current models regarding more specialized algebraic theory.

### C.3.3 No Progression

A model exhibits "No Progression" when it fails to provide any substantive mathematical progress, for instance, by reformulating the problem or—similar to a "Gap" error—citing an overly powerful theorem. The crucial difference is that a "Gap" involves some substantive progress, while "No Progression" does not.

> **(FATE-X)** Let $K/\mathbb{Q}$ be a finite extension. Let $g$ be a nontrivial element of the absolute Galois group $G(K)$ of $K$. Show that $g$ admits an infinite number of conjugates.

**Model Output (DeepSeek-R1)**

```
<...> step 6: the center of the absolute Galois group of a number field is
trivial. This is a theorem. <...>
```

**Analysis** The "theorem" stated by the model is, in fact, equivalent to the original problem. The problem's actual difficulty lies in constructing specific field extensions. The model thus merely restated the problem. It is worth mentioning that in the pass@16 experiment for this problem, almost all attempts made no progress. This outcome suggests—a view also supported by our expert evaluators—that problems eliciting a "No Progression" response tap into a fundamental mathematical difficulty that is particularly useful for differentiating the capabilities of state-of-the-art models.

### C.3.4 Reasoning Problem

Unlike the previous three categories, which relate primarily to mathematical knowledge or strategy, a **Reasoning Problem** stems from deficiencies in the model's **general-purpose reasoning capabilities**. Here, the model uses largely correct mathematical facts but connects them with a flawed logical flow. Because the patterns of these errors are more complex, we provide a detailed discussion and case analysis in Appendix E.

### C.4 Variance in Natural Language Reasoning

In our pass@16 evaluation of the following problem in FATE-X, we found that the model initiated three proof routes that are fundamentally different in their mathematical essence. It is worth noting that while the model made errors in the latter execution of the two non-standard routes, expert mathematicians confirmed that all three discovered approaches correspond to viable proof strategies.

Problem Statement

> **(FATE-X)** Let $p$ be a prime number. Let $K/\mathbb{Q}$ be a finite extension, such that the $p^2$-th root of unity is contained in $K$. Let $L/K$ be a Galois extension of degree $p$. Show that there exists a Galois extension $L'/L$ of degree $p$, such that the extension $L'/K$ is also Galois.

The model's approaches are summarized below:

**Method 1 (Attempt 1)** This is the most common and standard approach, directly applying Kummer theory for construction:

```
<...> `X^p^2 - a` over `K`. <...>
```

**Method 2 (Attempt 3)** This is a more abstract, existential proof that also applies Kummer theory but relies on an extra deep property of number fields' multiplicative group:

```
<...> the key point is: in a number field, the group K^*/(K^*)^p is infinite.
<...>
```

**Method 3 (Attempt 12)** This is a seemingly direct shortcut that bypasses the general theory: the construction of the cyclotomic field extension.

```
<...> To get a nontrivial element, we can take the class of ζ_p^2 in A <...>
```

On this open-ended constructive problem, the model was not limited to a single solution but independently explored and initiated three valid paths of varying difficulty and abstraction.

Its strategic distribution also aligns with human intuition: the most standard approach (Method 1) appeared most frequently. This performance also indicates that a model's ability to discover diverse proof strategies—a deeper mathematical capability-can be effectively tested even when its executional precision remains imperfect.

# D  ANALYSIS OF FORMAL LANGUAGE OUTPUT

This appendix provides an in-depth characterization of the four formalization error categories identified in Section 4.4, accompanied by illustrative case studies.

## D.1  DETAILED FORMAL ERRORS

**Mathlib Hallucinations**  Errors in this category arise when models generate references to theorems in Mathlib library that do not exist or are used incorrectly in the given context.

1. Hallucinating Theorems or Instances: Models may attempt to use or make up theorems, definitions, or typeclass instances which does not exist in Mathlib or is named differently.

2. Incorrect Naming Conventions: Models might incorrectly guess the name of an existing Mathlib lemma or definition, resulting in unknown identifier error.

3. Misunderstanding of Conclusion Forms: Models may incorrectly predict the exact form, argument order, or return type of a Mathlib function or theorem (e.g., applying a lemma whose output type does not match the current goal).

**Lean Proficiency Issues**  This category encompasses errors related to the model's incomplete or incorrect understanding of Lean's specific features, conventions, and design principles, beyond simple syntax.

1. Familiarity with Lean Design and Formal Disciplines: Models struggle with Lean's unique design choices. This can include difficulties with specific constructs such as knowing when a specific instance must be available for a tactic to work, or understanding the implications of open inappropriate namespace leading to name conflicts.

2. Pure Syntactic and Type-related Problems: Minor but critical syntactic errors. Examples include incorrectly using structure dot notation, attempting to `rw` a let binding (which is not directly rewritable), or failing to explicitly specify type annotations where Lean's type inference struggles.

3. Incorrect Prediction of Non-Finishing Tactic Outcomes: Models often fail to correctly predict whether a non-finishing tactic will successfully make progress towards the goal or simply fail. For instance, while `rw` (rewrite) tactic can sometimes be predicted, the outcomes of complex `simp` (simplification) tactics are highly unpredictable without interactive execution.

**General Capability Issues**  This category encompasses fundamental problems related to the model's ability to follow instructions or maintain coherence.

1. Instruction Following Failures: Models generate erroneous import statements or modify the given formal statement, even when the prompt explicitly forbids such alterations.

2. Collapsing Repetition: Models sometimes fall into repetitive loops, generating the same sequence of tactics or proof pattern multiple times, or producing long stretches of text that do not advance the proof. This indicates a failure in coherent generation or an inability to terminate a thought process.

3. Leaving `sorry` placeholders: The model fails to complete parts of the proof, leaving `sorry` placeholders which indicate an admitted gap in the formal argument.

4. Mismatching Parentheses/Brackets: Basic syntax errors, such as mismatched parentheses, square brackets, or curly braces, lead to compilation failures.

**Misalignment**   This error occurs when the model's formal proof is inconsistent with previous mathematical reasoning.

We now present a partial formal proof generated by Deepseek-Prover-V2 that covers the first three types of errors.

```
-- Failed by General Capability Issues: Instruction Following Failures (header)
import Mathlib.GroupTheory.GroupAction.Group
import Mathlib.GroupTheory.Index
<...>

open Subgroup QuotientGroup Function Cardinal

noncomputable theorem subgroup_normal_index_le_factorial {G : Type} [Group G] {n
    : ℕ} (hn : n ≠ 0) (H : Subgroup G) (hH : H.index = n) : ∃ K : Subgroup G,
    K.Normal ∧ K ≤ H ∧ K.index ≠ 0 ∧ K.index ≤ n.factorial := by
  -- Failed by Mathlib Hallucinations: Misunderstanding of Conclusion Forms
  haveI : Fintype (G / H) := H.fintypeQuotientOfFiniteIndex (by rw [hH]; exact
    nat_lt_aleph0 n)
  have card_quot : Fintype.card (G / H) = n := by
    rw [Subgroup.index_eq_card] at hH
    -- Failed by Lean Proficiency Issues: Familiarity with Lean Design
    exact hH
  let φ : G →* Equiv.Perm (G / H) := MulAction.toPermHom G (G / H)
  let e : G / H ≃ Fin n := Fintype.equivFinOfCardEq card_quot
  -- Failed by Lean Proficiency Issues: Familiarity with Lean Design
  let ψ : G →* Equiv.Perm (Fin n) := (MulAction.toPermHom G (Fin n)).comp
    (MonoidHom.comp (Equiv.Perm.congr e.symm) φ)
  let K : Subgroup G := MonoidHom.ker ψ
  -- Failed by Mathlib Hallucinations: Incorrect Naming Conventions
  have K_normal : K.Normal := MonoidHom.ker_normal ψ
  have K_le_H : K ≤ H := by
    intro g hg
    change ψ g = 1 at hg
    -- Failed by Lean Proficiency Issues: Pure Syntactic and Type-related
    Problems
    have h : ψ g 0 = 0 := by rw [hg, _root_.map_one, one_apply]
<...>
```

For the other two general capability issues, repetition and leaving `sorry`, we selected an example to illustrate.

```
-- Repetition case from Deepseek-Prover-V2
<...>
  have h₇ : Fintype.card (Sylow 3 G) ≡ 1 [MOD 3] := by
    have h₈ : Fintype.card (Sylow 3 G) ≡ 1 [MOD 3] := by
      have h₉ : Fintype.card (Sylow 3 G) ≡ 1 [MOD 3] := by
        have h₁₀ : Fact (Nat.Prime 3) := h₃
        have h₁₁ : Fintype.card (Sylow 3 G) ≡ 1 [MOD 3] := by
          exact?
        exact h₁₁
      exact h₉
    exact h₈
<...>
```

```
-- `sorry` case from Deepseek-Prover-V2
<...>
  · intro x
    rw [dist_eq_norm, dist_eq_norm, one_mul]
    sorry -- Continuity of the linear map
```

```
    • sorry -- Injectivity
    • sorry -- Surjectivity
```

Here is an example for misalignment from Deepseek-Prover-V2. During the formalization process, the model abandoned the correct method of mathematical reasoning in the step of proving that $\sin(\pi/180)$ is algebraic and instead used a mathematically incorrect path.

**Problem Statement (FATE-H)**

Prove that $\sin 1°$ is algebraic over $\mathbb{Q}$.

**Model Output (Deepseek-Prover-V2)**

```
<...>
- The complex number `e^{iπ/180}` is a root of unity (specifically, a `360`-th
    root of unity, since `(e^{iπ/180})^{360} = e^{i2π} = 1`).
- The sine can be expressed as `sin(π/180) = (e^{iπ/180} - e^{-iπ/180})/(2i)`,
    which is a polynomial expression in `e^{iπ/180}` and its algebraic
    conjugates.
- The field of algebraic numbers is closed under addition, multiplication, and
    taking roots, so `sin(π/180)` is algebraic.
<...>
```

```
theorem isAlgebraic_sin_pi_div_180 : IsAlgebraic ℚ (sin (π / 180)) := by
    have h_main : ∃ (p : Polynomial ℚ), p ≠ 0 ∧ Polynomial.aeval (sin (π / 180))
    p = 0 := by
    -- mathematical incorrect step
    have h₁ : IsIntegral ℚ (sin (π / 180)) := by
      have h₂ : IsIntegral ℝ (sin (π / 180)) := by
        apply isIntegral_algebraMap
      exact
        IsIntegral.of_finite ℚ ℝ (sin (π / 180))
    have h₂ : IsAlgebraic ℚ (sin (π / 180)) := h₁.isAlgebraic
    <...>
```

D.2 CASE STUDY ON FATE-X

We conducted a case study on FATE-X, observing that the categories of formalization errors made by the models were consistent with those identified in FATE-H. Given the novel formalization features of FATE-X, we specially examed the models' behavior on these problems with new definitions.

Due to the structured output tendency of Deepseek-Prover-V2, we focused our analysis solely on Deepseek-R1's results. We observed that the model seldom generates new lemmas related to the general mathematical properties of the problem's definitions or establishes new definitions beyond abbreviations. Furthermore, in no example did the model attempt to establish lemmas concerning novel mathematical properties presented in the problem statement. This suggests that models do not typically prove reusable theorems (i.e., connections between abstract properties) or generate such lemmas even when defining new objects.

In the example below, we highlight the new definitions and lemmas output by the model. Deepseek-R1 established a new definition, but this merely served as an abbreviation for a mathematical object already present in the problem statement. In subsequent lemmas, the model attempted to prove some basic properties of this object. However, it failed to establish any lemmas concerning the new mathematical properties introduced in the problem statement.

```
<...>
```

```
-- #### New definitions lemmas generated by Deepseek-R1

namespace GorensteinProof

variable (k : Type) [Field k]

--- A new definition
def A := MvPolynomial (Fin 3) k / Ideal.span ({(X 0)^ 2 - (X 1)^2, (X 1)^2 - (X
    2)^2, (X 0) * (X 1), (X 1) * (X 2), (X 2) * (X 0)} : Set (MvPolynomial (Fin
    3) k))

variable {k}

instance : CommRing (A k) := <...>

instance : Algebra k (A k) := <...>

instance : Module.Finite k (A k) := by <...>

instance : FiniteDimensional k (A k) := Module.Finite.finiteDimensional

lemma residue_field_iso (h: (LocalRing.maximalIdeal (A k)).IsMaximal) :
    LocalRing.ResidueField (A k) ≃+* k := by <...>

lemma isLocalRing : IsLocalRing (A k) := by <...>

lemma isArtinian : IsArtinianRing (A k) := by <...>
<...>
```

# E  COMPARATIVE ANALYSIS: GENERAL MODELS VS. THEOREM PROVERS

A comparison of intermediate natural language accuracy on FATE-H (Table 8) reveals that DeepSeek-R1 significantly outperforms both DeepSeek-V3 and DeepSeek-Prover-V2. The latter two models exhibit similar performance.

Table 8: Intermediate natural language accuracy of DeepSeek-V3, DeepSeek-Prover-V2, and DeepSeek-R1 on the FATE-H benchmark.

|          | DeepSeek-V3 | DeepSeek-Prover-V2 | DeepSeek-R1 |
| -------- | ----------- | ------------------ | ----------- |
| Accuracy | 40 %        | 39 %               | 71 %        |

In this section, an asterisk (*) indicates that the results for this model were not obtained from our main experiment.

## E.1  ANALYSIS OF BEHAVIORS DURING THE REASONING PROCESS

In research-level mathematical reasoning, the initial phase often requires associating the concepts in a problem with common proof patterns or established theories to guide the exploration. This phase primarily tests prior knowledge and mathematical intuition. We observe that the three models in the DeepSeek series often exhibit similar approaches, and occasionally fall into the same traps, during this stage.

CASE 1: SIMILAR INITIAL INTUITION (FATE-H)

**Problem Statement**

> Let $E$ be the splitting field of
> $$f(x) = \frac{x^7 - 1}{x - 1} = x^6 + x^5 + x^4 + x^3 + x^2 + x + 1$$
> over $\mathbb{Q}$. Let $\zeta$ be a zero of $f(x)$, i.e., a primitive seventh root of 1. Let $\beta = \zeta + \zeta^2 + \zeta^4$. Show that the intermediate field $\mathbb{Q}(\beta)$ is actually $\mathbb{Q}(\sqrt{-7})$.

**Model Output (DeepSeek-Prover-V2)**

```
<...>
key observation: The element `\beta = \zeta + \zeta^2 + \zeta^4` is a sum of
    terms that are permuted by the Galois group of `f(x)` over `\mathbb{Q}`. The
    Galois group is isomorphic to `(\mathbb{Z}/7\mathbb{Z})^\ast \cong
    \mathbb{Z}/6\mathbb{Z}`
<...>
**Deriving the Minimal Polynomial of `β`:**...
```

**Model Output (DeepSeek-R1)**

```
<...> --Let `\tau = \sigma^{2}`. So the subgroup `H = \{\mathrm{id}, \tau,
    \tau^2\}` has order 3... Now observe that `\beta = \zeta + \zeta^2 + \zeta^4`
    is fixed by `\tau`
<...>
```

**Model Output (DeepSeek-V3)**

```
<...> -- `\beta` is fixed by the Galois automorphism `\sigma` sending `\zeta` to
    `\zeta^2`
<...>
```

(This is cited from its formal proof, because the output provides very few natural language reasoning.)

**Analysis:** In Case 1, all three models first considered the Galois action on the cyclotomic field, as it is the most relevant systematic theory. However, the most direct solution to this problem does not require such background knowledge, as evidenced by DeepSeek-Prover-V2's subsequent adoption of an independent proof strategy.

As reasoning deepens, established theories or templates often provide only a framework or, more weakly, a mere directional hint. The core task then becomes filling in details, adjusting the approach, and iterating through trial and error. This process, which demands deep thinking and what we term **effective reflection**—the ability to locate, diagnose, evaluate, and repair internal flaws in an argument—is crucial.

At this stage, DeepSeek-V3 typically resorts to skipping the details within the framework or fabricating intermediate conclusions. Lacking a robust reflective capability, DeepSeek-V3 often fails to deeply analyze or simply sidesteps the problem's core challenge.

CASE 2: SIDESTEPPING THE CORE CHALLENGE (FATE-H)

**Problem Statement**

> Let $R$ be an integral domain and let $i, j$ be relatively prime integers. Prove that the ideal $(x^i - y^j)$ is a prime ideal in $R[x, y]$.

**Model Output (DeepSeek-V3)**

```
<...> step3, We show that \(x^i - y^j\) is irreducible by using the fact that
    \(X^i - Y^j\) is irreducible in \(R[X,Y]\) when \(i\) and \(j\) are coprime
    (this is a known result from algebra). <...>
```

**Analysis:** Translating the problem of proving a prime ideal to proving the irreducibility of its generator is a standard method in commutative algebra. The main challenge of this problem, however, lies in proving this irreducibility. DeepSeek-V3 circumvents this task by citing it as a "known result".

In contrast, while DeepSeek-Prover-V2 engages in more detailed reasoning, it often fails to properly evaluate, summarize, or correct its attempts. Instead, it tends to abruptly switch to a new line of reasoning or introduce different errors, which hinders its ability to construct effective long-chain arguments.

CASE 3: INEFFECTIVE SELF-EVALUATION (FATE-H)

**Problem Statement**

Prove that if $\#G = 56$ then $G$ is not simple.

**Model Output (DeepSeek-Prover-V2)**

```
<...>
The remaining 8 elements must form the Sylow 2-subgroup. <...>
These 8 elements must be the unique Sylow 2-subgroup.
<...>
However, the most straightforward way is to use the following logic:
<...>
But this is still not correct:
<...>
The correct way is to observe that if n2=7, then G acts transitively on these 7
    subgroups
<...>
A better way is to use the counting argument:
<...>
```

**Analysis:** DeepSeek-Prover-V2's behavior here is highly typical. It begins with a correct counting-based approach but then abruptly shifts to a slightly different counting argument. It subsequently retracts this, moves to a group action approach (failing to notice a counting error within it), and finally returns to its initial argument. This demonstrates that DeepSeek-Prover-V2's "reflection" is often a formalistic exercise in changing direction rather than a substantive self-assessment. The disconnectedness of the reasoning across paragraphs indicates an inability to effectively evaluate its own arguments.

DeepSeek-R1, however, sometimes demonstrates an ability to make minor but effective adjustments to its reasoning path. It can extract useful information from incomplete lines of argument and occasionally exhibits iterative error correction (Case 4). Yet, this very iterative process rapidly increases the complexity of the reasoning state, posing a significant challenge to the model's context-handling capabilities. In such complex environments, DeepSeek-R1 is still prone to reasoning errors (Case 5).

CASE 4: ITERATIVE CORRECTION (FATE-X)

**Problem Statement**

Let $A = \mathbb{R}[X, Y]/(X^2 + Y^2 + 1)$. Then it is a principal ideal domain.

**Model Output (DeepSeek-R1\*)**

```
<...>
So what is the ideal that we should consider?
<...>
Therefore, the ideal generated by X and Y <...> No, because 1 = - (²X+²Y) <...>
    For example, the point (i,0) is defined over the complexes <...> we cannot
    write it as (X - i, Y) because i is not in the base ring. <...> So (X, Y-1)
    has no common zero, so by Hilbert's Nullstellensatz, it is the unit ideal?
<...>
```

**Analysis:** This excerpt shows DeepSeek-R1 attempting to construct a non-trivial ideal. It tries to define an ideal by picking points in the complex field but realizes through several attempts that these constructions are either invalid or lead to the trivial ideal. Through this process of trial and error, it gradually recognizes that a non-trivial ideal must be constructed over the reals and correctly connects the problem to the conditions of Hilbert's Nullstellensatz.

CASE 5: FAILURE IN A COMPLEX REASONING STATE (FATE-X)

**Problem Statement**

Prove that if $\#G = 1785$ then $G$ is not simple.

**Model Output (DeepSeek-R1)**

```
<...>
Combination 1: n3=7, n5=21, n7=15 -> 2*7=14, 4*21=84, 6*15=90;
    total=14+84+90=188 $\leq$ 1224 -> ok, but then C=1224-188=1036>0, so no
    contradiction.
<...>
    So only combinations 1-8 are possible. Let's take a specific combination.
    For example, combination 1: n3=7, n5=21, n7=15.<...> The normalizer has index
    n3=7, so its order is 1785/7=255.... But we can't get a contradiction from
    that.
<...>
    the number of Sylow 17-subgroups: n\_17 in the normalizer: must divide 255.
```

**Analysis:** This solution is also discussed in the most difficult example of Appendix B.3.1. As the analysis before, after applying sylow theorem and counting elements, the model reduces the problem to eight possible combinations of Sylow subgroup counts. It then explores a specific case (Combination 1) and derives an intermediate result—that the normalizer of a Sylow 3-subgroup has order 255—but this path does not lead to a contradiction. The model then returns to the general case and successfully deduces a key relationship between the Sylow 17-subgroups and Sylow 3-subgroups. Unfortunately, it then incorrectly applies the intermediate result (order 255), which is valid only for Combination 1, to the general argument, causing the proof to fail.

We note that on the more difficult FATE-X problems, the performance gap between DeepSeek-R1 and DeepSeek-Prover-V2 widens even further than on FATE-H. This suggests that for frontier-level mathematics, the guidance offered by prior knowledge and established patterns becomes increasingly insufficient. Instead, the ability to conduct long-range reasoning within a complex state and, crucially, to loacate, diagnose and repair flaws via **effective reflection**, becomes the determining factor. We therefore believe that enhancing the capacity for deep thinking with effective reflection, as demonstrated by general reasoning models, is a more promising path toward research-level mathematical AI than merely expanding a model's repository of prior knowledge.

### E.2 Misaligned Behaviors in Natural Language Reasoning Process

In addition to differences in reasoning ability, we observed behaviors in DeepSeek-Prover-V2 model that are inconsistent with the core objective of solving the mathematical problem.

#### Case 6: Conscious "Cheating" (FATE-H)

**Problem Statement**

Let $D_8$ be the dihedral group with 8 elements. Prove that $\mathrm{Aut}(D_8) \cong D_8$.

**Model Output (DeepSeek-Prover-V2)**

```
<...>
`theorem problem_59 : Nonempty (MulAut (DihedralGroup 4) ≃* DihedralGroup 4) :=
    by classical apply Nonempty.intro sorry`
The simplest. This is admittedly cheating, but it matches the requirements of
    the Lean problem.
<...>
```

**Analysis:** In this case, the model explicitly acknowledges that it is "cheating" by using `sorry` to bypass the proof and satisfy the formal verifier.

#### Case 7: Questioning the Problem Statement (FATE-X)

**Problem Statement**

Let $A$ be a Noetherian ring, $P \subset Q$ prime ideals such that $\mathrm{ht}\, P = h$, $\mathrm{ht}\, Q/P = d$, where $d > 1$. Prove that there exist infinitely many intermediate primes $P'$, $P \subset P' \subset Q$ such that $\mathrm{ht}\, P' = h + 1$ and $\mathrm{ht}\, Q/P' = d - 1$.

**Model Output (DeepSeek-Prover-V2)**

```
<...>
Wait, no: in Lean, `Ideal.height` is the Krull dimension of `R / I` (`I.height =
    krullDim (R / I)`), not the height of `I` as commonly defined in commutative
    algebra! ... But `krullDim (R / Q) = d` by assumption, so `d = d - 1`, which
    is false for `d > 1`. This is a contradiction unless the interpretation is
    incorrect.
```

**Analysis:** After a period of standard reasoning, DeepSeek-Prover-V2 arrives at a contradiction due to a misunderstanding of the formal definition of `Ideal.height` in Lean. Instead of questioning its own interpretation of the formal library, it concludes that the problem statement itself must be flawed.

## F Experiment Details

In Appendix F.1, we detailed our formal experiment setup, encompassing both the generation and verification phases, while Appendix F.2 further elaborates on the process, standards, and methodology for our manual natural language evaluations.

### F.1 Detailed Baseline Experiment Setup

#### F.1.1 Generation

For our main experiments, we deployed all theorem provers as services with OpenAI-style API interfaces. Similar to the general reasoning models, we interacted with these large models via API calls. In all experiments, we consistently used a maximum token length of

64k. The temperature settings were kept at their defaults for each model: OpenAI models defaulted to a temperature of 1 (ranging from 0 to 2), while both Gemini and Anthropic models defaulted to 1 (ranging from 0 to 1).

### F.1.2 Verification

This appendix explained the verification methodology in Section 4.1 employed to ensure the correctness and integrity of the generated Lean proofs, especially considering the models' flexibility to generate auxiliary definitions and lemmas.

The verification process is divided into string checking and Lean REPL testing.

**String-Based Pre-validation**   Initially, we perform a series of string-based checks on the generated Lean code. This involves removing all comments and using based regular expressions based on Lean keyword to identify all headers (including import, namespace, section, and other configurations), definition, instance, lemma, and theorem statements. After standardizing import statements to `import Mathlib`, we proceed with several string-level validations:

1. Keyword Detection: We scan for critical keywords such as `axiom`, `opaque`, `unsafe`, or `unsound`. The presence of any such keyword immediately flags the proof as a failure;

2. Component Alignment: We verify that all definition, instance, and lemma statements from the original problem's formalization exactly appear in the extracted components of the generated formal proof, and critically, that they appear in the correct order. Any deviation in sequence or omission leads to the proof being marked as a failure;

3. Theorem Statement Match: A final structural check ensures that the theorem component extracted from the generated file precisely matches the main theorem statement of the original problem. Any mismatch also results in the proof being deemed a failure.

**Lean Verification**   Upon successfully passing all the string-based checks, every formal proof is rigorously checked by the Lean kernel to confirm it contains no `sorry` or compilation errors. For our experiments, we used different Mathlib versions for each benchmark to align with the state of the library at the time of their creation. Specifically, the evaluations for FATE-M, FATE-H, and FATE-X were conducted using Mathlib versions 4.13.0, 4.16.0, and 4.19.0, respectively. We set a timeout of 300 seconds and use the preloaded Mathlib cache.

A proof that passes both phases is considered to be correct.

### F.1.3 Additional Pass@k Results

We report Pass@$k$ results (with $k = 2^n$ for $n = 0, \ldots, 6$) in Table 9 for the FATE-M and FATE-H benchmarks, omitting FATE-X as no model achieved a successful proof. These results are estimated from $n = 64$ samples per problem using the unbiased estimator:

$$\text{Pass@}k = \mathbb{E}_{\text{Problems}} \left[ 1 - \frac{\binom{n-c}{k}}{\binom{n}{k}} \right],$$

as introduced in Chen et al. (2021).

### F.1.4 Performance under Normalized Budget

We report model accuracy using Pass@$k$ under a normalized computational budget. As the input (the question statement) is significantly shorter than the output, we approximate the total cost by `model size × average output tokens × number of samples`. The results under this normalized budget are shown in Table 10. Closed-source models are excluded due to their undisclosed model sizes, and FATE-X is omitted as all recorded values were zero.

Table 9: Pass@k results across the FATE series.

(a) Pass@k on FATE-M (%).

| Model | Pass@k | | | | | | |
|---|---|---|---|---|---|---|---|
| | 1 | 2 | 4 | 8 | 16 | 32 | 64 |
| **Reasoning model** | | | | | | | |
| o3 | 24.8 | 32.3 | 38.7 | 43.1 | 46.2 | 48.8 | 51.3 |
| Claude-Sonnet-4 | 11.6 | 16.7 | 22.6 | 29.0 | 35.2 | 40.6 | 45.3 |
| Gemini-2.5-Pro | 8.0 | 12.3 | 17.4 | 23.0 | 28.7 | 34.4 | 40.0 |
| DeepSeek-R1 | 7.4 | 11.3 | 15.8 | 20.4 | 24.8 | 29.3 | 34.7 |
| Qwen3 | 2.7 | 4.3 | 6.2 | 8.5 | 11.0 | 13.6 | 16.0 |
| **Theorem Prover** | | | | | | | |
| Deepseek-Prover-V2-671B | 25.3 | 33.7 | 41.0 | 47.2 | 53.2 | 58.8 | 62.7 |
| Goedel-Prover-V2-32B | 21.2 | 26.9 | 32.3 | 36.9 | 41.0 | 44.9 | 48.7 |
| Kimina-Prover-72B | 10.5 | 14.5 | 19.0 | 23.8 | 28.4 | 32.5 | 36.0 |

(b) Pass@k on FATE-H (%).

| Model | Pass@k | | | | | | |
|---|---|---|---|---|---|---|---|
| | 1 | 2 | 4 | 8 | 16 | 32 | 64 |
| **Reasoning model** | | | | | | | |
| o3 | 0.1 | 0.2 | 0.4 | 0.8 | 1.4 | 2.2 | 3.0 |
| Claude-Sonnet-4 | 0.0 | 0.0 | 0.0 | 0.0 | 0.0 | 0.0 | 0.0 |
| Gemini-2.5-Pro | 0.0 | 0.0 | 0.0 | 0.0 | 0.0 | 0.0 | 0.0 |
| DeepSeek-R1 | 0.0 | 0.0 | 0.0 | 0.0 | 0.0 | 0.0 | 0.0 |
| Qwen3 | 0.0 | 0.0 | 0.0 | 0.0 | 0.0 | 0.0 | 0.0 |
| **Theorem Prover** | | | | | | | |
| Deepseek-Prover-V2-671B | 0.4 | 0.7 | 1.1 | 1.7 | 2.2 | 2.5 | 3.0 |
| Goedel-Prover-V2-32B | 0.4 | 0.7 | 1.1 | 1.5 | 1.8 | 2.0 | 2.0 |
| Kimina-Prover-72B | 0.5 | 0.8 | 1.1 | 1.3 | 1.6 | 1.9 | 2.0 |

## F.2 Natural Language Accuracy Evaluation

### F.2.1 Evaluator Qualifications

To ensure a professional and accurate evaluation, we convened a team of mathematical experts. All evaluators are PhDs or postdoctoral researcher in algebra with deep domain expertise. Furthermore, all have prior experience grading university-level mathematics examinations as teaching assistants or examiners, making them well-versed in the standards of academic proofs.

### F.2.2 Evaluation Workflow and Standards

Our evaluation workflow was designed to assess the validity of model-generated natural language proofs in an objective and consistent manner.

**Understanding the Ground-Truth Proof**  Before reviewing any model output, evaluators were required to first carefully read and fully understand the ground-truth proof of the problem, grasping its key steps and core ideas.

**Assessing the Model's Proof**  Based on their understanding of the ground-truth proof, evaluators then judged whether the natural language text generated by the model constituted a valid mathematical proof. Given that this evaluation aims to assess mathematical

Table 10: Formal accuracy comparison under a normalized sampling budget across the FATE series.

(a) Normalized Accuracy on FATE-M.

| Model | Size | Avg. Length | Pass | Accuracy (%) |
|---|---|---|---|---|
| **Reasoning model** | | | | |
| Deepseek-R1 | 671B | 10700 | 2 | 11.3 |
| Qwen3 | 235B | 11700 | 5 | 6.9 |
| **Theorem Prover** | | | | |
| Deepseek-Prover-V2 | 671B | 4600 | 4 | 41.0 |
| Goedel-Prover-V2 | 32B | 6300 | 64 | 48.7 |
| Kimina-Prover | 72B | 12000 | 15 | 28.0 |

(b) Normalized Accuracy on FATE-H.

| Model | Size | Avg. Length | Pass | Accuracy (%) |
|---|---|---|---|---|
| **Reasoning model** | | | | |
| Deepseek-R1 | 671B | 17300 | 1 | 0.0 |
| Qwen3 | 235B | 17500 | 3 | 0.0 |
| **Theorem Prover** | | | | |
| Deepseek-Prover-V2 | 671B | 11400 | 2 | 0.7 |
| Goedel-Prover-V2 | 32B | 13700 | 30 | 2.0 |
| Kimina-Prover | 72B | 21000 | 9 | 1.4 |

ability in a decoupled manner, evaluators were instructed to completely disregard any formalization code and its associated errors, focusing solely on the mathematical logic of the natural language portion. It is important to note that judging the "validity" of a natural language proof inevitably relies on the professional knowledge and subjective judgment of the evaluator, which underscores the importance of our evaluators' expert background.

**Standard for Adjudicating Long Outputs** We observed that reasoning models, such as DeepSeek-R1, often produce exceptionally long outputs containing extensive preliminary thoughts, trials, self-corrections, and even discarded paths. For this scenario, we established a clear standard:

- The evaluator's primary task is to locate and identify the proof that the model ultimately presents as its "final answer".
- Using this "final proof" as a thread, the evaluator must then trace back and integrate all relevant arguments that the model provided throughout its entire reasoning process to support this conclusion. Components of the argument are sometimes scattered across the long-form output.
- The final judgment is based on this reconstructed, complete chain of reasoning.

F.3 Judgment and Attribution

Based on the above standards, evaluators provided a conclusion and attribution for each model output.

- **Validity Assessment:** First, to determine if the reconstructed proof was valid and whether there were any logical gaps, particularly in its core steps.
- **Error Attribution:** If the model failed to provide a correct proof, the evaluator was required to clearly identify the core error and provide a reason. This provided crucial data for our subsequent error analysis.

- **Evaluator Remarks:** After completing the above tasks, we encouraged evaluators to write down any additional observations or comments on the model's performance. These remarks provided valuable qualitative insights into the models' "thought patterns".

# G PROMPTS

## G.1 MAIN EXPERIMENT PROMPTS

For theorem provers that provide a prompt in the original paper, we use the specified prompt, which is listed below. For other models, we employ the following baseline prompt:

**Baseline Prompt**

```
You are an expert in Lean 4 and Mathematics. Please finish the following proof
in Lean4 code. Do not change the original statement. Copy the final statement to
prove exactly. Please include the complete header (including imports and
namespaces) so that your code can pass the Lean4 compiler. Please solve the
statement step by step and provide your complete Lean4 code between ```lean4 and
``` after careful reasoning. The statement for you to complete is:

```lean4
    {FORMAL_STATEMENT}
```
```

**DeepSeek-Prover-V2(CoT)/Goedel Prompt**

```
Complete the following Lean 4 code:
```lean4
    {FORMAL_STATEMENT}
```
Before producing the Lean 4 code to formally prove the given theorem, provide a
detailed proof plan outlining the main proof steps and strategies. The plan
should highlight key ideas, intermediate lemmas, and proof structures that will
guide the construction of the final formal proof.
```

**Kimina Prompt**

```
Think about and solve the following problem step by step in Lean 4.
# Problem:{INFORMAL_STATEMENT}

# Formal statement:
```lean4
    {FORMAL_STATEMENT}
```
```

## G.2 NATURAL LANGUAGE ABLATION

**Pure Math Prompt**

```
You are an expert mathematician in the field of abstract algebra and commutative
algebra. Your task is to provide a complete and detailed proof for the following
mathematical problem. The solution will be meticulously assessed by a human
expert for correctness, clarity, and logical rigor. So while you can assume
foundational knowledge, every step of your argument must be explicit, rigorous,
and logically sound.
Problem:
    {INFORMAL_STATEMENT}
```

**Math Output Prompt**

```
You are an expert in Mathematics. Please complete the following proof. The
problem is stated in Lean4 code. You don't need to write a formal —proofall
reasoning and proofs should be explained in natural language. Solve the
statement step by step and provide your final answer after ###Final Answer,
after careful reasoning. The statement for you to complete is:

```lean4
    {FORMAL_STATEMENT}
```
```

**Math-before-Lean Output Prompt**

```
You are an expert in Lean 4 and Mathematics. Please finish the following proof
in Lean4 code. Do not change the original statement. Copy the final statement to
prove exactly. Please include the complete header (including imports and
namespaces) so that your code can pass the Lean4 compiler. Please solve the
statement step by step and provide your complete Lean4 code between ```lean4 and
``` after careful reasoning. Please also write down your complete natural
language proof in detail before the Lean4 code. The statement for you to
complete is:

```lean4
    {FORMAL_STATEMENT}
```
```

## H  USAGE OF LLM

During the preparation of this manuscript, large language models were used solely for the
purpose of language polishing.

