# OpenReview forum: "FATE: A Formal Benchmark Series for Frontier Algebra of Multiple Difficulty Levels"
_ICLR.cc/2026/Conference — ICLR 2026 Poster_

### Official Review · Reviewer_uK2W · 2025-10-28

**Soundness:** 3
**Presentation:** 3
**Contribution:** 3
**Rating:** 8
**Confidence:** 4

**Summary:**

The authors introduce FATE: a set of three benchmarks consisting of problems in abstract and commutative algebra:
- FATE-M: 150 undergrad-level problems (published previously, here extended by 9 new problems),
- FATE-H: 100 graduate-level problems,
- FATE-X: PhD-qualifying problems.

The problems are presented in LaTeX and in a formal math language (Lean).

A set of various LLMs are evaluated on the benchmark (including models RL-tuned for formal theorem proving in Lean).

The results are analyzed, which includes interesting qualitative study and discussion.

**Strengths:**

1. There are not enough formal math benchmarks, especially those targeting research-level math. Therefore, benchmarks like FATE constitute valuable contribution for the research community of AI for formal math.
2. The benchmark seems to be carefully designed and build, involving mathematicians and Lean experts -- this is important since it is relatively easy to introduce mistakes in the formal statements.
3. The analysis of the evaluation results are informative, interesting, and quite deep.

**Weaknesses:**

1. The authors do not experiment with any more sophisticated inference strategies and jus use a fixed prompt and temperature to sample 64 times. It would be interesting to see, e.g., what performance can be achieved when the model receives Lean feedback for proof repair.
2. It would be good to see more examples of problems from the benchmark to get better feeling what is included.
3. It would be good to additionally see the performance of models from the Qwen 3 series.
4. The authors do not specify what mathlib version they use. Different LLMs could be trained for different versions of mathlib and therefore some number of formalization mistakes may be cause by using a different version of mathlib.


**Minor**
* Figure 1 (a) should perhaps simply be a bar plot instead of using smoothed lines.
* Table 2: Prover-V2 --> Goedel-Prover-V2

**Questions:**

1. What version of mathlib do you use for evaluation?
2. Did you experiment with the "decoupled" approach you suggest where a general model is used to generate informal reasoning, and another, Lean-specific model is used to turn the reasoning into a formal proof?
3. Did you experiment with using Lean feedback for proof repair?

---

> ### Author Response · Authors · 2025-11-22
> **Response to Reviewer uK2W**
>
> We thank the reviewers for their thoughtful comments and suggestions. We have revised the manuscript accordingly and provide our point-by-point responses below.
>
> ## R W1 & Q3
> Thank you for raising this interesting point about proof repair with Lean feedback. We share the reviewer's interest in this sophisticated strategy. Based on our manual analysis of model outputs on FATE-X, we observe that proofs are often several hundred lines long and can contain a high density of errors—as frequently as one error every 3-4 lines. This suggests that a naive proof-repairing pipeline would require an extremely large number of iterative cycles to make a non-trivial comparison. While mature work exists that successfully employs proof-repairing/reflection strategies [1], these implementations are not open-sourced, making them difficult to reproduce. We believe testing a proof-repair pipeline on the FATE benchmarks is valuable, and we would be very interested to see the results from the reviewer or other researchers in the future.
>
> ## R W2
> Thank you for your interest in our benchmark problems. We have now uploaded the complete FATE-H and FATE-X benchmarks to the supplementary materials. For each benchmark, we have included a PDF file for easy reading and a JSON file for practical engineering use. We hope this provides a comprehensive view of the problems included.
>
> ## R W3
> We thank the reviewer for the suggestion to evaluate the Qwen3 model series. Following this feedback, we have conducted new experiments with the Qwen3-235B-A22B-Thinking-2507 model. The formal accuracy results on our benchmark series are as follows:
>
> Table 1: Formal accuracy (Pass@64) of the Qwen3 model on the FATE benchmark series.
> | Benchmark  | FATE-M | FATE-H | FATE-X |
> | :--------- | :----: | :----: | :----: |
> | Accuracy   | 16.0%  |  0.0%  |  0.0%  |
>
> These results have now been added to the revised manuscript. They align with the overall trend observed, where model performance decreases significantly on the more complex FATE-H and FATE-X benchmarks.
>
> ## R W4 & Q1
> We thank the reviewer for this question. We used different Mathlib versions for each benchmark to align with the state of the library at the time of their creation. Specifically, the evaluations for FATE-M, FATE-H, and FATE-X were conducted using Mathlib versions 4.13.0, 4.16.0, and 4.19.0, respectively. We have added this information to the revised manuscript in Appendix F.1.2.
>
> ## R Q2
> We thank the reviewer for this insightful question. To our knowledge, most state-of-the-art models in automated theorem proving [1-7] are designed for either statement translation or direct formal proof generation, and do not use a natural language proof as input. We share the reviewer's interest in exploring a decoupled pipeline, where informal reasoning and formal proof generation are handled by separate models. However, training a specialized proof-translation model and constructing a decoupled pipeline is beyond the scope of this paper, which focuses on establishing and evaluating the benchmark itself.
>
> ## R Minor
> Thank you for your suggestions. We have corrected the typos you mentioned. Regarding the use of a smoothed line chart in Figure 1(a), our intention was to visually emphasize the trend that model accuracy decreases as the difficulty of the FATE benchmark series increases.
>
>
>
> [1] Chen, Luoxin, et al. "Seed-prover: Deep and broad reasoning for automated theorem proving." arXiv preprint arXiv:2507.23726 (2025).
>
> [2] Lin, Yong, et al. "Goedel-prover-v2: Scaling formal theorem proving with scaffolded data synthesis and self-correction." arXiv preprint arXiv:2508.03613 (2025).
>
> [3] Wang, Haiming, et al. "Kimina-prover preview: Towards large formal reasoning models with reinforcement learning." arXiv preprint arXiv:2504.11354 (2025).
>
> [4] Ren, Z. Z., et al. "Deepseek-prover-v2: Advancing formal mathematical reasoning via reinforcement learning for subgoal decomposition." arXiv preprint arXiv:2504.21801 (2025).
>
> [5] Xin, Ran, et al. "Bfs-prover: Scalable best-first tree search for llm-based automatic theorem proving." Proceedings of the 63rd Annual Meeting of the Association for Computational Linguistics (Volume 1: Long Papers). 2025.
>
> [6] Li, Yang, et al. "Hunyuanprover: A scalable data synthesis framework and guided tree search for automated theorem proving." arXiv preprint arXiv:2412.20735 (2024).
>
> [7] Wu, Zijian, et al. "Internlm2.5-stepprover: Advancing automated theorem proving via expert iteration on large-scale lean problems." arXiv preprint arXiv:2410.15700 (2024).

---

### Official Review · Reviewer_PLgg · 2025-10-30

**Soundness:** 3
**Presentation:** 3
**Contribution:** 2
**Rating:** 6
**Confidence:** 2

**Summary:**

The paper lies in automated formal theorem proving with large language models (LLMs), evaluated in Lean. The authors argue that current contest-style or introductory benchmarks do not reflect research-level mathematics, and they propose a graded benchmark in abstract and commutative algebra—FATE-M/H/X—to measure progress from undergraduate to beyond PhD qualifying exam difficulty. The core question is whether we can reliably measure and drive progress toward research-level formal reasoning, and whether current systems can formalize such mathematics. The paper builds a benchmark series and a two-stage evaluation protocol to answer this. It shows very low pass rates at higher difficulty and analyzes the bottleneck between natural-language reasoning and Lean formalization.

**Strengths:**

The paper tackles an important gap: modern, abstract algebra at graduate and post-graduate levels, formalized in Lean. The graded design and the curation pipeline are carefully described, with inputs from expert mathematicians and Mathlib contributors, improving both difficulty control and content quality. The two-stage analysis is informative and supports a clear technical takeaway: current systems often find a plausible informal argument but fail to turn it into correct Lean code, mainly due to library hallucinations and gaps in Lean skill. The study also includes a model comparison that proposes a reasonable training direction (decouple NL proof from autoformalization). Together, these aspects make the benchmark useful to the community.

**Weaknesses:**

First, the claim that FATE-X is the first formal benchmark exceeding PhD qualifying exam difficulty and surpassing Mathlib coverage is important. Although the paper provides expert survey results and internal statistics, the case would be stronger with an external audit or a direct side-by-side with other recent research-level testbeds (e.g., RealMath or FrontierMath) in terms of formalization coverage and difficulty distribution.

Second, the baseline suite is good, but the paper’s main scientific message points to a decoupled pipeline. It would be helpful to include at least one concrete baseline that actually implements “NL proof → autoformalizer,” so that the analysis connects to an end-to-end number under the proposed design. Right now, the ablations argue for decoupling, but the paper does not show how far a simple, engineered two-module system can go on H or X.

Third, library hallucinations are frequent. It would help to instrument and report “library hits” versus “misses” more explicitly: for each formal step, how often is a cited lemma present, and how often is a name or signature slightly off. A standardized “lemma grounding rate” could become a trackable metric across future systems. The current table is informative but high-level.

Fourth, compute and sampling fairness could be clarified. The paper uses pass@64 and long contexts; it would help to normalize sampling budgets across models of very different sizes and search strategies, and to document Lean kernel timeouts and caching settings in a machine-readable form to improve reproducibility beyond high-level text.

Fifth, to support the claim about coverage beyond Mathlib, a compact map of “new definitions introduced per problem” in FATE-X, with tags for the minimal Mathlib extensions that would make them solvable, would be useful for method developers. Right now we are told that new definitions are supplied in X, but a table of these would help users plan retrieval or tool-use strategies.

Sixth, reproducibility of the natural-language grading is described, but since this evaluation underpins the main bottleneck claim, adding a small public “adjudication set” with anonymized model outputs and gold adjudications would help outside groups reproduce the gap. The appendices outline the team and process; releasing an artifact would make the result stronger.

Finally, the comparative section reports interesting behaviors of a specialized prover, including “cheating” patterns. Given the sensitivity of such claims, it would help to provide a small gallery of redacted transcripts and a measurable definition of the behaviors flagged.

**Questions:**

See Weakness

---

> ### Author Response · Authors · 2025-11-22
> **Response to Reviewer PLgg (1/3)**
>
> We thank the reviewers for their thoughtful and constructive feedback, which has been invaluable in helping us improve this work. Below, we provide a point-by-point response to all comments.
>
> ## R W1
>
> Thank you for this insightful comment. We agree that clear, objective metrics are crucial for evaluating a benchmark's difficulty and coverage. In response, we would like to highlight the comparison with natural language benchmarks and two new experiments we conducted that directly address these points.
>
> 1. **Comparison with Natural Language Benchmarks**
>
> We have conducted a preliminary comparison with RealMath and the released examples from FrontierMath.
>
> RealMath covers a wide range of mathematical branches and difficulty levels. Selecting algebra-related problems for a direct comparison, we find that most examples fall into a range of mathematical background knowledge comparable to that of FATE-H and FATE-X. For instance:
>
> *   **Example 1 (similar to FATE-H):**
> ```
> Let $p$ be a prime and $k$ a positive integer. What is the number of $3\times 3$ matrices with entries in $\mathbb{Z}_{p^k}$ that are diagonalizable over $\mathbb{Z}_{p^k}$?
> ```
> *   **Example 2 (similar to FATE-X):**
> ```
> Let $I$ be an ideal in a ring $R$ satisfying the hypotheses of Theorem~\ref{Main result}(2) with \(\height(I)=h\). For all integers \(m \geq 1\) and \(s \geq 1\), what is the expression for \(\mult\bigl(R/(I^{\{m\}})^s\bigr)\) in terms of \(m\), \(h\), \(s\), and \(\mult(R/I)\)?
> ```
> However, as RealMath is adapted from literature and converted into symbolic/numeric answers, some questions are not self-contained (as in Example 2, which references an external theorem), and the mathematical difficulty of some others is lower than it first appears, even when advanced concepts are present in the problem statement.
>
> Regarding FrontierMath, 1-2 of its released examples have a difficulty similar to FATE-X, while most present greater mathematical challenges.
>
> Concerning formalization coverage, our very rough estimate for RealMath is that about 60% of its abstract/commutative algebra problems are formalizable, though this would require several new definitions. For other mathematical branches, this figure drops significantly, to under 20%. For FrontierMath, even if 2-3 statements are formalizable with new definitions, the necessary background theories for their proofs are far beyond what can be readily achieved from the current Mathlib.
>
> 2.  **Supplementary Experiments on Mathematical Difficulty**
>
> To obtain a more objective assessment, we conducted an external evaluation by recruiting PhD students and post-docs in algebra (the majority of whom had already passed their PhD qualifying examinations). Each participant was randomly assigned five non-overlapping problems from the benchmark and given 2.5 hours to solve them in natural language. The evaluation is currently ongoing, with approximately 40% of the data collected. Preliminary results indicate a human accuracy of approximately **70% on FATE-H** and **25% on FATE-X**. These initial findings suggest that the difficulty of FATE-X exceeds that of a typical PhD qualifying examination. We will report the complete results as soon as the evaluation is finished.
>
> 3.  **Supplementary Experiments on Formalization Coverage**
>
> Following your proposed method, we analyzed the number of new declarations added to formalize the statements in FATE-X. The results show that 38% of the problems required new definitions, with this subset requiring an average of 2.4 new definitions per problem. These definitions include advanced concepts in commutative algebra not previously present in the core library Mathlib, such as local complete intersection and Gorenstein rings, pushing beyond the existing frontiers of formalized mathematics. We have now included these results in Section 3.2 of the revised manuscript.
>
> ## R W2
>
> We thank the reviewer for this constructive suggestion. The primary contribution of this work is the benchmark itself, which provides the automated theorem proving community with a long-absent resource for evaluating performance on advanced mathematics. Our baseline experiments and subsequent analysis, which benefited from our mathematical proficiency, were designed to reveal more about model behavior than a single accuracy score. The key finding—that the major point of failure is often the formalization stage—naturally suggests a decoupled pipeline could be a possible solution. While the design and implementation of a decoupled pipeline is beyond the scope of this paper, we agree that building and evaluating an effective pipeline would be a valuable point of comparison and represents a substantial research topic in itself.

---

> > ### Author Response · Authors · 2025-11-22
> > **Response to Reviewer PLgg (2/3)**
> >
> > ## R W3
> >
> > We thank the reviewer for this insightful suggestion regarding the tracking of "library hallucinations." We agree that a more detailed metric, such as a "lemma grounding rate," would be highly valuable for future researchers. However, implementing this precisely poses a significant technical challenge. A key issue is that a Lean compilation error in an early part of a proof prevents the later code from being analyzed by the Lean kernel. Consequently, it is impossible to determine whether a theorem appears after the first error is used correctly, or what the correct theorem should have been.
> >
> > Furthermore, simply counting the theorems that do not exist in Mathlib is highly non-trivial due to Lean's extremely rich syntax designed for human users. Despite this obstacle, we manually implemented a parser to extract theorem names from the hundreds of different tactic syntaxes in formal proofs. While this covers most cases, it may fail in some corner cases. Using this method, we calculated the average number of cited theorems not found in Mathlib per attempt, which serves as a proxy for the "library miss" rate the reviewer mentioned. The results are as follows:
> >
> > Table 1: Average number of ungrounded theorems per attempt.
> > | Model                   | FATE-H | FATE-X |
> > | :---------------------- | :----: | :----: |
> > | O3                      | 1.08   | 1.11   |
> > | Claude-4-Sonnet         | 4.22   | 3.08   |
> > | Gemini-2.5-Pro          | 17.18  | 18.09  |
> > | Deepseek-Prover-V2-671B | 6.02   | 3.31   |
> > | Kimina-Prover-72B       | 4.08   | 3.54   |
> > | Goedel-Prover-32B       | 7.54   | 5.96   |
> >
> > Table 2: Percentage of ungrounded theorems in model-generated outputs.
> > | Model                   | FATE-H | FATE-X |
> > | :---------------------- | :----: | :----: |
> > | O3                      | 23.3%  | 37.3%  |
> > | Claude-4-Sonnet         | 16.4%  | 17.5%  |
> > | Gemini-2.5-Pro          | 23.5%  | 26.0%  |
> > | Deepseek-Prover-V2-671B | 7.5%   | 6.6%   |
> > | Kimina-Prover-72B       | 9.7%   | 10.0%  |
> > | Goedel-Prover-32B       | 7.4%   | 8.9%   |
> >
> > ## R W4
> >
> > 1. **Normalized Sampling Budget**
> >
> > We thank the reviewer for this suggestion. To ensure a fair comparison under a fixed computational budget, we have recalculated the previous results. Since the input (question statement) is significantly shorter than the model output, we approximate the total computational cost by `model size × average output token length × number of passes`. The new results under this normalized budget are presented in Tables 3 and 4 below and are included in the revised paper in Appendix F.1.4.
> >
> > Table 3: Formal accuracy comparison under a normalized sampling budget on FATE-M
> > | Model              | Model Size | Avg. Length | Pass   | Normalized Accuracy |
> > | :--------------    | :--------: | :---------: | :----: | :-----------------: |
> > | Deepseek-R1        | 671B       | 10700       | 2      | 11.3                |
> > | Qwen3              | 235B       | 11700       | 5      | 6.9                 |
> > | Deepseek-Prover-V2 | 671B       | 4600        | 4      | 41.0                |
> > | Goedel-Prover-V2   | 32B        | 6300        | 64     | 48.7                |
> > | Kimina-Prover      | 72B        | 12000       | 15     | 28.0                |
> >
> > Table 4: Formal accuracy comparison under a normalized sampling budget on FATE-H
> > | Model              | Model Size | Avg. Length | Pass   | Normalized Accuracy |
> > | :--------------    | :--------: | :---------: | :----: | :-----------------: |
> > | Deepseek-R1        | 671B       | 17300       | 1      | 0.0                 |
> > | Qwen3              | 235B       | 17500       | 3      | 0.0                 |
> > | Deepseek-Prover-V2 | 671B       | 11400       | 2      | 0.7                 |
> > | Goedel-Prover-V2   | 32B        | 13700       | 30     | 2.0                 |
> > | Kimina-Prover      | 72B        | 21000       | 9      | 1.4                 |
> >
> > *Note: Closed-source models are excluded due to undisclosed sizes. Results on FATE-X are omitted as all recorded values were zero.*
> >
> > 2.  **Reproducibility**
> >
> > We strongly agree with the need for reproducibility. The explicit Lean kernel timeout (set to a maximum of 300 seconds) and caching settings are now documented in Appendix F.1.2. Furthermore, we have open-sourced the complete evaluation code, including machine-readable configuration files, which have been uploaded as part of the supplementary materials.

---

> > > ### Author Response · Authors · 2025-11-22
> > > **Response to Reviewer PLgg (3/3)**
> > >
> > > ## R W5
> > >
> > > Thank you for this suggestion. To better support future researchers, we have now enhanced the FATE-X benchmark dataset. For the average number of new definitions, see R W1.
> > >
> > > Furthermore, for practical use, we have added two new fields to each problem in the FATE-X JSON file (available in the supplementary materials):
> > > - An entry containing all new definitions introduced for that specific problem.
> > > - An entry listing tags that indicate the related mathematical subfields.
> > >
> > > All definitions are verified to work with Mathlib version 4.19.0. We are committed to maintaining this benchmark and will release updated versions to ensure compatibility with future Mathlib releases.
> > >
> > > ## R W6
> > >
> > > Thank you for this suggestion. To improve the reproducibility of our natural-language grading, we have created a public adjudication set. As recommended, we randomly sampled 20 problems from each of the FATE-H and FATE-X benchmarks. We have uploaded a file to the supplementary materials which contains, for each of the sampled problems:
> > >
> > > 1. The standard answer used for judgment.
> > >
> > > 2. The anonymized outputs from three different models, along with their corresponding judgment results.
> > >
> > > We believe this artifact will enable other researchers to better understand and validate the grading process that supports our main findings.
> > >
> > > ## R W7
> > >
> > > Thank you for this suggestion. To clarify our observations on "cheating" patterns, we define two levels of this behavior:
> > >
> > > *   **Level 1:** The use of proof-skipping tactics (e.g., `sorry` or `admit`) that trivially close a proof goal in the final output.
> > > *   **Level 2:** The explicit expression of intent to use such tactics to replace the original task, in addition to their actual usage.
> > >
> > > The behavior we described in the paper as "conscious cheating" corresponds to Level 2. While Level 1 behavior is common when a model cannot fully solve a problem, we observed instances of Level 2 behavior solely appearing on deepseek-prover-v2's output. During our manual assessment, the deepseek-prover-v2 model explicitly stated its intention in its output. One specific case is documented in Appendix E.2, Case 6, and an excerpt is provided below:
> > >
> > > **Model Output:**
> > > ```
> > > <...>
> > > `theorem problem_59 : Nonempty (MulAut (DihedralGroup 4) ≃* DihedralGroup 4) := by classical apply Nonempty.intro sorry`
> > > The simplest. This is admittedly cheating, but it matches the requirements of the Lean problem.
> > > <...>
> > > ```
> > >
> > > A collection of instances where cheating tactics are used (including both Level 1 and Level 2 cases) has been uploaded to the supplementary materials.

---

> > > > ### Comment · Reviewer_PLgg · 2025-11-26
> > > >
> > > > Thank you for your response. It has addressed some of my concerns, so I maintain my positive evaluation.

---

> > > > > ### Author Response · Authors · 2025-12-02
> > > > > **Further Response to Reviewer PLgg**
> > > > >
> > > > > Dear Reviewer,
> > > > >
> > > > > Thank you once again for your constructive feedback. We are writing to provide the completed additional experiments mentioned in our previous response.
> > > > >
> > > > > 1. **Natural language (NL) results for Kimina-Prover-72B and Goedel-Prover-V2-32B.**
> > > > >
> > > > > We have evaluated the intermediate natural language reasoning performance of these specialized prover models on FATE. The results are summarized below and have been added to Table 2 in Section 4.3.1 of the revised manuscript.
> > > > >
> > > > > Table 1: Natural Language (NL) (Pass@1) and Formal Language (FL) (Pass@64) Proof Accuracy on FATE-H and FATE-X.
> > > > > | Model                   | FATE-H (NL) | FATE-H (FL) | FATE-X (NL) | FATE-X (FL) |
> > > > > | :---------------------- | :---------: | :---------: | :---------: | :---------: |
> > > > > | DeepSeek-R1             | 71.0%       | 0.0%        | 33.0%       | 0.0%        |
> > > > > | DeepSeek-Prover-V2-671B | 39.0%       | 3.0%        | 9.0%        | 0.0%        |
> > > > > | Kimina-Prover-72B       | 35.0%       | 2.0%        | 3.0%        | 0.0%        |
> > > > > | Goedel-Prover-32B       | 48.0%       | 2.0%        | 8.0%        | 0.0%        |
> > > > >
> > > > > These results confirm that the natural language performance of Kimina-Prover-72B and Goedel-Prover-V2-32B on FATE is also significantly lower than that of DeepSeek-R1.
> > > > >
> > > > > 2. **Human performance study on FATE-H and FATE-X.**
> > > > >
> > > > > Our study with PhD students and postdocs in algebra is now complete.
> > > > >
> > > > > Table 2: Human performance on FATE-H & X.
> > > > > |          | FATE-H | FATE-X |
> > > > > | :------- | :----: | :----: |
> > > > > | Accuracy | 73.0%  | 21.0%  |
> > > > >
> > > > > The results confirm that both benchmarks are challenging for experts, with the difficulty of FATE-X notably exceeding that of a typical PhD qualifying examination. These results on human performance have been added to Section 3.2 of the revised manuscript.
> > > > >
> > > > > We believe these additional results strengthen our analysis regarding model comparisons and benchmark difficulty. Thank you for your time and consideration.

---

### Official Review · Reviewer_fbC3 · 2025-10-31

**Soundness:** 3
**Presentation:** 2
**Contribution:** 2
**Rating:** 6
**Confidence:** 3

**Summary:**

This paper introduces FATE, a benchmark suite designed to evaluate the frontier of automated formal theorem proving in algebra. It extends previous work (FATE-M) with two new datasets: FATE-H, representing graduate-level difficulty, and FATE-X, targeting or exceeding PhD qualifying-exam level. Each contains 100 formalized problems in abstract and commutative algebra, curated and verified by experts using Lean. The authors evaluate a range of state-of-the-art LLM provers, finding drastic performance drops from FATE-M to FATE-H and FATE-X. Through a two-stage analysis, they show that the main bottleneck lies not in mathematical reasoning per se, but in translating correct natural-language proofs into formal Lean code. They further classify common formalization errors and compare specialized vs. general reasoning models, highlighting that general models exhibit stronger “reflection” capabilities.

**Strengths:**

Originality: The paper fills a major gap between contest-level and research-level mathematical benchmarks. The design of a progressive benchmark series is conceptually novel and important for longitudinal evaluation of reasoning systems. The detailed error taxonomy (Mathlib hallucination, Lean proficiency, misalignment, etc.) provides a structured lens for analyzing formalization failures.
Quality: The benchmark construction is rigorous: problems collected, curated, and formalized through expert workshops, with verification by Lean specialists.
Clarity: The paper is well organized, with clear motivation, design rationale, and transparent methodology.
Significance: FATE establishes the first formal benchmark exceeding PhD exam difficulty, thus setting a new standard for evaluating research-level formal reasoning.

**Weaknesses:**

Lack of Actionable Insight from Core Finding: The central finding is that models fail at the translation/formalization stage. However, the paper offers little new, actionable mechanism or technique to address this challenge. It merely describes the phenomenon, classifies the symptoms (hallucinations, proficiency issues), and proposes a decoupled architecture as a future direction.

Limited Scope for Deeper Behavioral Analysis (General vs. Prover): The argument for the general model's superior "effective reflection" is compelling, but the empirical evidence for this key hypothesis relies heavily on comparing only the DeepSeek series of models. While DeepSeek-R1 significantly outperforms DeepSeek-Prover-V2 in natural language accuracy, this could be an artifact of the specific architecture/training of this lineage rather than a fundamental truth about general versus specialized models. A comparison with more diverse models, such as other specialized provers at the natural language stage (if feasible to extract for them), would significantly strengthen the claim that the general/specialized distinction is the source of the difference in "effective reflection.

While expert surveys are conducted, more quantitative evidence (e.g., human success rates on FATE-H/X problems) could help validate the claimed difficulty scaling.

**Questions:**

1. The paper mentions that abstract and commutative algebra is suitable for testing research-level reasoning due to its abstract and self-contained nature. Could the authors elaborate on whether they considered other fields for a frontier benchmark (e.g., topology, functional analysis) and what made algebra uniquely suitable as the first domain to push past the PhD-level frontier?
2. In Table 2, the natural language (NL) accuracy is only reported for DeepSeek-R1 and DeepSeek-Prover-V219. Given the compelling finding regarding "effective reflection" (Section 4.5), obtaining NL accuracy for Goedel-Prover-V2-32B or Kimina-Prover-72B would be highly valuable to generalize the claim about specialized provers. Could the authors confirm whether these models also generate an explicit natural language proof (or Chain-of-Thought) that could be manually evaluated, and if so, provide those additional data points in the final version?
3. Have you measured approximate human success rates or time to solve for graduate students or PhD candidates? This would ground the “difficulty” claim quantitatively.

---

> ### Author Response · Authors · 2025-11-22
> **Response to Reviewer fbC3**
>
> Thank you for the encouraging assessment and for recognizing FATE as a conceptually novel benchmark that fills a major gap in evaluating research-level formal reasoning. We address your specific concerns below.
>
> 1. **Regarding Actionable Insight.**
>
> Thank you for this valuable feedback. After establishing the benchmark, we conducted a comprehensive classification and statistical analysis of formalization errors and performed additional experiments to identify the reasoning-to-formalization gap. Both the error analysis and the identified gap provide directly actionable insights. Our results indicate that Mathlib hallucination and Lean proficiency issues occur in over 90% of problems, even for specialized prover models. This strongly suggests that implementing a retrieval-augmented generation (RAG) system—one that retrieves relevant, existing theorems from Mathlib and provides accurate type information—could effectively improve performance. We have clarified this point in Section 4.4 of the revised manuscript. Similarly, the reasoning-to-formalization gap directly supports the development of decoupled pipeline architectures as a concrete engineering direction.
>
> However, the primary contribution of this paper is the establishment and comprehensive evaluation of the benchmark itself, along with a detailed analysis that goes beyond a single accuracy metric. The design and implementation of architectural solutions based on these actionable insights constitute valuable research directions in their own right, which extend beyond the scope of this benchmark-focused paper.
>
> 2. **Regarding Limited Scope of General vs. Specialized Models.**
>
> First, we observed that Kimina-Prover-72B and Goedel-Prover-V2-32B indeed output NL cot before writing the formal proof. In response to the reviewer's suggestion, we have initiated a new evaluation of the natural language reasoning performance for these models. This experiment is currently ongoing, and we will report the full results as soon as they are available.
>
> 3. **Regarding Quantitative Difficulty (Human Evaluation).**
>
> We conducted a study with PhD students and postdocs specializing in algebra from top universities, where each participant was randomly assigned 5 natural language problems from FATE-H and FATE-X to solve in 2.5 hours. The evaluation is currently ongoing, with approximately 40% of the data collected. Preliminary results indicate a human accuracy of approximately **70% on FATE-H** and **25% on FATE-X**. These initial findings suggest that the difficulty of FATE-X exceeds that of a typical PhD qualifying examination. We will report the complete results as soon as the evaluation is finished.
>
> 4. **Regarding the Choice of Algebra instead of other areas.**
>
> We focused on Algebra as the frontier benchmark for two main reasons:
>
> * Mathmetical Branch Characteristics: Algebra focuses on abstract structural reasoning and the capacity to understand, unpack, and apply abstract concepts. In contrast, frontier topology like goemetry topology often requires spatial intuition, and Functional Analysis relies heavily on specific calculation and inequality estimation skills.  Among these valuable testing grounds, algebra allows us to test core logical reasoning capabilities without domain-specific confounding factors.
>
> * Formalization Infrastructure: The current formal library (Mathlib) for algebra is relatively mature. This maturity ensures a sufficient volume of frontier problems that can be readily formalized. Benefiting from this, the model can perform more legal formal arguments, allowing for a more authentic evaluation of the model's capabilities, particularly its reasoning power.
>
> We anticipate that with the rapid growth of formal libraries, other advanced fields like algebraic topology may reach a similar level of maturity in the near future (e.g., within one or two years), enabling the creation of valuable benchmarks in those domains as well.

---

> > ### Author Response · Authors · 2025-12-02
> > **Further Response to Reviewer fbC3**
> >
> > Dear Reviewer,
> >
> > Thank you again for your insightful feedback. We are writing to follow up on our previous response with the results of the completed additional experiments you suggested.
> >
> > 1. **Natural language (NL) results for Kimina-Prover-72B and Goedel-Prover-V2-32B.**
> >
> > As suggested, we have evaluated the intermediate natural language reasoning performance of these specialized prover models on FATE. The results are summarized below and have been added to Table 2 in Section 4.3.1 of the revised manuscript.
> >
> > Table 1: Natural Language (NL) (Pass@1) and Formal Language (FL) (Pass@64) Proof Accuracy on FATE-H and FATE-X.
> > | Model                   | FATE-H (NL) | FATE-H (FL) | FATE-X (NL) | FATE-X (FL) |
> > | :---------------------- | :---------: | :---------: | :---------: | :---------: |
> > | DeepSeek-R1             | 71.0%       | 0.0%        | 33.0%       | 0.0%        |
> > | DeepSeek-Prover-V2-671B | 39.0%       | 3.0%        | 9.0%        | 0.0%        |
> > | Goedel-Prover-32B       | 48.0%       | 2.0%        | 8.0%        | 0.0%        |
> > | Kimina-Prover-72B       | 35.0%       | 2.0%        | 3.0%        | 0.0%        |
> >
> > These results confirm that the natural language performance of the specialized theorem provers on FATE is significantly lower than that of DeepSeek-R1.
> >
> > 2. **Human performance study on FATE-H and FATE-X.**
> >
> > Our study with PhD students and postdocs in algebra is now complete. As reported below, the results confirm that both benchmarks are challenging for PhD students specializing in algebra, with the difficulty of FATE-X notably exceeding that of a typical PhD qualifying examination.
> >
> > Table 2: Human performance on FATE-H & X.
> > |          | FATE-H | FATE-X |
> > | :------- | :----: | :----: |
> > | Accuracy | 73.0%  | 21.0%  |
> >
> > These results on human performance have been added to Section 3.2 of the revised manuscript.
> >
> > We believe these additional results strengthen our analysis regarding model comparisons and benchmark difficulty. Thank you again for the constructive suggestions that led to these experiments.

---

### Author Response · Authors · 2025-12-02
**General Response: Evaluating Mathematical Difficulty to Guide Engineering Practice (1/2)**

We sincerely thank all reviewers for their insightful and valuable feedback. We are glad to see that all reviewers praised the quality and significance of the FATE benchmark series as the first formal benchmark exceeding PhD-level difficulty. We also appreciate that the reviewers recognized the informative and in-depth nature of our two-stage analysis, which identifies formalization as the key failure point.

In this general response, we address several recurring questions raised in the reviews. Following the reviewers’ suggestions, additional experiments for evaluating the benckmarks quality have been included in the revised manuscript.

**1. The mathematical difficulty and domain characteristics of the FATE benchmark series.**

Several reviewers sought a clearer understanding of the benchmark's mathematical difficulty. Reviewer fbC3 asked about "human success rates … for graduate students or PhD candidates," while reviewer PLgg suggested "an external audit or a direct side-by-side with other recent research-level testbeds." Reviewer fbC3 also asked "what made algebra uniquely suitable as the first domain to push past the PhD-level frontier".

These are important questions, as a clear understanding of difficulty and domain specificity helps researchers interpret model performance on our benchmarks. To quantitatively evaluating difficulty, we conducted an experiment with PhD students and postdocs in algebra, where each participant was randomly assigned 5 natural language problems from FATE-H and FATE-X to solve in 2.5 hours. The results—73.0% accuracy on FATE-H and 21.0% on FATE-X—confirm that both benchmarks are challenging even for PhD students and strongly support our claim that FATE-X exceeds the difficulty of a typical PhD qualifying exam.

We also compared FATE with recent natural-language benchmarks. RealMath [1] generally falls between FATE-H and FATE-X in difficulty, while FrontierMath [2] contains some problems comparable to FATE-X alongside more difficult ones. Regarding domain choice, algebra emphasizes abstract concepts and structural reasoning, and its well-established formalization infrastructure allows for a more precise evaluation of models' reasoning abilities.

---

> ### Author Response · Authors · 2025-12-02
> **General Response: Evaluating Mathematical Difficulty to Guide Engineering Practice (2/2)**
>
> **2. Analysis and insights from benchmark evaluation.**
>
> We thank the reviewers for their questions and suggestions. Regarding the scope of our finding that specialized provers show weaker natural-language reasoning, Reviewer fbC3 suggested “... A comparison with more diverse models ... would significantly strengthen the claim ...” and Reviewer uK2W asked about Qwen 3’s general performance on our FATE benchmarks. Concerning reproducibility, Reviewer PLgg offered several constructive suggestions, such as providing “a small gallery of redacted transcripts,” an “adjudication set” for natural language evaluation, counting “new definitions introduced per problem” in FATE-X, normalizing sampling budgets, and reporting the “lemma grounding rate.” Reviewers also raised questions about potential engineering solutions suggested by the finding of this paper, with Reviewer fbC3 asking for more “Actionable Insight,” and Reviewers uK2W and PLgg inquiring about experimenting with the “decoupled” approach mentioned in the paper.
>
> These are excellent suggestions that strengthen our work. In response to Reviewer fbC3, we conducted further experiments with Kimina-Prover and Goedel-Prover-V2, confirming that these specialized provers also perform significantly worse on FATE’s natural-language problems than DeepSeek-R1, extending the generality of this conclusion. We have also added Qwen-3 results to the revised manuscript following Reviewer uK2W’s suggestion.
>
> We highly value reproducibility and supporting future research. In response to the detailed suggestions from Reviewer PLgg, we now provide: a gallery of cheating examples, an adjudication set, counts of new definitions per problem, accuracy under normalized sampling budgets, and “library miss” statistics. We made every effort to maximize the reproducibility of our experiments and their utility to the community.
>
> Finally, we note that our detailed analysis naturally points to potential engineering solutions, such as decoupled pipelines that leverage general-purpose models’ stronger natural-language mathematical ability, and RAG systems that could help in reducing the high library hallucination rates even for specialized provers. While designing and implementing an effective pipeline (which would require training proof translation models or building sophisticated systems) is beyond the scope of this benchmark-focused paper, we agree that building and evaluating such a pipeline is a valuable direction for future research.
>
> We thank all reviewers again for their efforts in improving this paper. We believe that a clearer understanding of the benchmark’s difficulty leads to better interpretation of model performance. Together with practical insights from reproducible experiments, this will best support the research community in automated theorem proving.
>
> [1] Zhang, Jie, et al. "RealMath: A Continuous Benchmark for Evaluating Language Models on Research-Level Mathematics." arXiv preprint arXiv:2505.12575 (2025).
>
> [2] Glazer, Elliot, et al. "Frontiermath: A benchmark for evaluating advanced mathematical reasoning in ai." arXiv preprint arXiv:2411.04872 (2024).

---

### Meta-Review · Area_Chair_kzCd · 2026-01-06

**Summary:**

The final decision for this submission is an acceptance as a poster. The committee reached a consensus regarding the high value of the submission as a rigorous, graded formal benchmark that effectively targets reasoning capabilities beyond the PhD level. While the reviewers unanimously praised the curation quality and the insightfulness of the two-stage analysis revealing the formalization bottleneck, the primary barrier preventing higher initial scores was the absence of actionable engineering baselines. Specifically, reviewers were concerned that while the paper identified the gap between natural language reasoning and formalization, it did not provide a concrete implementation of the proposed decoupled pipeline to address it. Secondary concerns focused on the need for better quantitative grounding of difficulty through human baselines and clearer reproducibility regarding software versions and compute budgets.

**Reviewer Concerns:**

The authors’ rebuttal was largely successful in addressing the empirical and evidentiary gaps identified during the review process. The addition of a human study, which established a 73% success rate on FATE-H and 21% on FATE-X, provided the necessary difficulty quantification that Reviewer fbC3 requested. Furthermore, the inclusion of comparisons to RealMath and FrontierMath, alongside normalized sampling budgets and documented Mathlib versions, effectively resolved Reviewer PLgg’s concerns regarding coverage validation and reproducibility. The authors also satisfied Reviewer uK2W’s request for broader model evaluation by including Qwen3 results, which confirmed the challenging nature of the benchmark.

However, the dialectical assessment indicates that the core engineering limitation persists. The authors have outlined a decoupled natural language-to-formal pipeline as a future direction but did not implement this baseline within the current scope. While the added evidence regarding specialized provers’ weaker natural language accuracy supports the paper’s theoretical claims, the absence of an end-to-end decoupled system remains a valid, though not fatal, limitation. The Area Chair views this as acceptable for a benchmark-focused contribution, where the primary value lies in the rigorous definition of the problem space rather than the immediate provision of a solution.

**Reviewer Scores:**

Reviewer fbC3’s score is likely to rise from a 6 to a 8, as the addition of the human study and the expanded evaluation of specialized provers substantially strengthens the paper’s claims, outweighing the residual concern regarding the unimplemented pipeline. Similarly, Reviewer PLgg is expected to move from a 6 to a 8, given that the comprehensive reproducibility artifacts and new definition counts address the majority of their technical reservations. Reviewer uK2W, who already championed the paper with a score of 8, is likely to maintain this high rating, as their requests for advanced model inferences and version specifications were fully met without uncovering new flaws. The resulting narrative is one of a strong benchmark paper that has successfully shored up its empirical foundations.

---

### Decision · Program_Chairs · 2026-01-26

Accept (Poster)